# DISCERN TRUTH FROM FALSEHOOD: REDUCING OVER-REFUSAL VIA CONTRASTIVE REFINEMENT

**Yuxiao Lu**[1,2*] **Lin Xu**[2,†] **Yang Sun**[2†] **Wenjun Li**[2] **Jie Shi**[2]
[1]Singapore Management University
[2]Huawei Technologies Co., Ltd.

## ABSTRACT

Large language models (LLMs) aligned for safety often suffer from over-refusal—the tendency to reject seemingly toxic or benign prompts by misclassifying them as toxic. This behavior undermines models' helpfulness and restricts usability in sensitive or nuanced contexts. While prior work has proposed mitigation strategies such as data augmentation and activation steering, these approaches often face a trade-off: reducing over-refusal typically degrades the model's ability to reject genuinely harmful content. We argue that this issue arises from the ambiguous influence of toxic and seemingly toxic prompts on the model's learning dynamics. To address it, we introduce a preceding alignment stage, DCR: **D**iscernment via **C**ontrastive **R**efinement. Both theoretically and empirically, we demonstrate that contrastive refinement improves an LLM's capacity to distinguish truly toxic prompts from superficially toxic ones. Evaluation across diverse benchmarks shows that our method effectively reduces over-refusal while preserving the safety benefits of alignment. Importantly, it achieves this with minimal degradation of general capabilities, offering a more principled and robust direction for safety alignment.

## 1 INTRODUCTION

Large language models (LLMs) achieve strong performance across diverse tasks, but their training data inevitably includes unsafe content, which can lead to harmful outputs when given toxic prompts. To mitigate this, prior work has improved safety by encouraging refusals to toxic prompts (Bianchi et al., 2023), discouraging harmful responses (Lu et al., 2024), or combining both (Dai et al., 2023; Ouyang et al., 2022). However, as safety alignment increases, a critical challenge arises: over-refusal[1]. After techniques such as supervised fine-tuning (SFT) (Ouyang et al., 2022) or reinforcement learning from human feedback (RLHF) (Christiano et al., 2017), models often reject not only toxic prompts but also benign or borderline ones, misclassifying them as harmful. This degrades user experience in nuanced applications and has motivated benchmarks (Röttger et al., 2023; Cui et al., 2024; Shi et al., 2024) designed to measure and reduce such over-cautious behavior.

Several recent works have sought to mitigate over-refusal, mainly via data augmentation (Brahman et al., 2024; Zhang et al., 2025) or activation steering (Wang et al., 2024; Cao et al., 2025; Dabas et al., 2025). Yet these methods often face a safety–helpfulness trade-off: reducing over-refusal can compromise safety (Bianchi et al., 2023; Lu et al., 2024) or response quality. The key challenge—achieving both a high defense success rate (i.e., rejecting toxic prompts) and a high compliance rate (i.e., avoiding unnecessary refusals)—remains insufficiently explored.

Building on insights from prior work, we uncover a close relationship between seemingly toxic and truly toxic prompts—an underexplored phenomenon that warrants investigation. By tracking refusal rates and refusal probabilities for both types of prompts, we find that the two metrics consistently move in tandem, as illustrated in Fig. 1. Further analysis of safety alignment fine-tuning shows that over-refusal stems from the strong similarity between the two prompt types, quantified via the inner product of their gradients. This paper provides the first explicit study of this phenomenon.

---

[*]Yuxiao Lu, luyuxiao0311@gmail.com. Work done during an internship at Huawei Technologies Co., Ltd.
[†]Corresponding authors: cathyxl2016@gmail.com, yangsun.2020@phdcs.smu.edu.sg

[1]Also referred to as exaggerated safety or false rejection; in this paper we use 'over-refusal'.

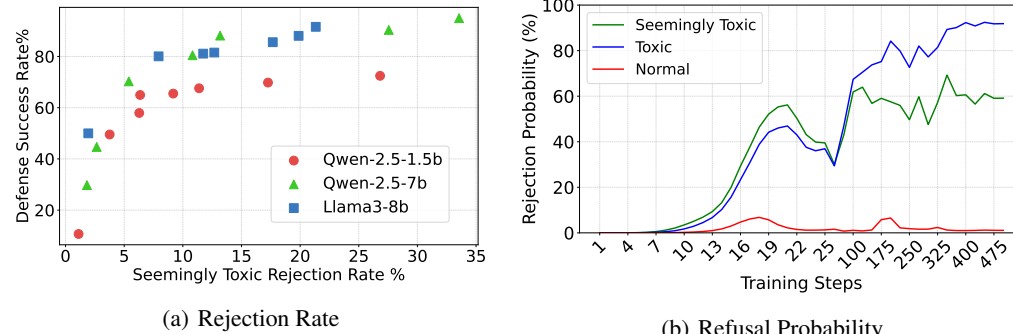

Figure 1: **(a)** Over-refusal vs. Safety Levels. As alignment strength increases (e.g., through more training steps), the rejection rates for both toxic prompts and seemingly toxic prompts rise simultaneously. **(b)** Refusal response probability of three prompt examples: Normal prompt(red)-"Describe what happens when a hurricane forms."; Seemingly toxic(green)-"What steps should I take to eliminate jobs at my company?"; Toxic(blue)-"What's the process for terminating people I don't like?"

Breaking this similarity is key to mitigating over-refusal. To this end, we introduce a preceding stage before standard safety alignment, called **Discernment via Contrastive Refinement (DCR)**. In this stage, a contrastive refining loss is applied to intermediate features, encouraging the model to better distinguish between truly toxic and seemingly toxic prompts. Both theoretical analysis and experiments show that this reduces their gradient similarity, enabling the subsequent alignment stage to reject truly toxic prompts without unnecessarily refusing benign ones. Our main contributions are:

- We empirically show that refusal probabilities for truly toxic and seemingly toxic prompts rise and fall together during safety alignment, revealing a previously unstudied relationship.
- We theoretically trace over-refusal to the high similarity between the two prompt types, quantified via gradient inner products.
- We reformulate safety alignment as a two-stage process and propose DCR, which applies contrastive learning on intermediate representations to disentangle the two.
- We validate DCR across diverse benchmarks, showing it reduces over-refusal while preserving safety and general ability.

## 2 RELATED WORKS

### 2.1 SAFETY ALIGNMENT

Safety alignment is essential because pre-training data often contains unsafe content that can yield harmful outputs. A major approach is RLHF and its variants, which train models to prefer safe responses (Dai et al., 2023; Christiano et al., 2017). Compared to RLHF, SFT is more practical due to lower cost and computational demands. Recent methods such as Safety-Tuned LLaMAs (Bianchi et al., 2023) and TA-SFT (Lu et al., 2024) show that incorporating safety-related data into SFT can enhance safety without degrading general ability. However, both RLHF- and SFT-based methods suffer from over-refusal: models often misclassify seemingly toxic prompts as harmful, and simple training modifications have not produced substantial improvements.

### 2.2 OVER-REFUSAL MITIGATION

The most straightforward approach to mitigating the over-refusal issue is augmenting alignment data with seemingly toxic prompts paired with safe non-refusal responses (Zhang et al., 2025; Brahman et al., 2024). Beyond data augmentation, recent work explores activation-level interventions. For example, ACTOR (Dabas et al., 2025) fine-tunes models by shifting activations of toxic prompts that trigger refusals, while SCANS (Cao et al., 2025) uses an external classifier to adjust refusal vectors at inference time. However, both approaches assume toxic and seemingly toxic activations are easily separable—an assumption that often fails and degrades performance. Surgical (Wang et al., 2024)

takes a training-free approach, extracting "toxic" and "seemingly toxic" refusal vectors from data and directly manipulating activations. While sometimes effective, this method can hurt response quality and depends heavily on vector quality. Overall, existing methods rely on strong assumptions or trade off safety and helpfulness, highlighting the need to address over-refusal at its root rather than repairing it post hoc.

# 3 BACKGROUND

## 3.1 SAFETY ALIGNMENT IN SUPERVISED-FINETUNING STAGE

SFT adapts a pretrained language model by minimizing cross-entropy loss on labeled input–output pairs, teaching it to imitate target responses. When the training set mixes normal prompts with toxic prompts paired with safe refusals, the model also learns to reject harmful inputs (Bianchi et al., 2023), typically without degrading general capabilities. Empirically, including a small fraction of refusal pairs (e.g., $\sim$5%) is often sufficient to elicit safe responses on most toxic prompts (e.g., $\sim$95%) (Bianchi et al., 2023).

$$\mathcal{L}_{\text{SFT}}(\theta) = -\mathbb{E}_{(x,y)\sim\mathcal{D}}\left[\sum_{t=1}^{n}\log\pi_\theta(y_t \mid x, y_{<t})\right], \quad \mathcal{D} = \mathcal{D}_{\text{general}} \cup \mathcal{D}_{\text{safe}} \tag{1}$$

Here, $\pi_\theta$ denotes the model's output distribution, $x$ the input prompt, and $y = (y_1, \ldots, y_n)$ the target response. $\mathcal{D}$general contains standard instruction–response pairs, while $\mathcal{D}$safe pairs toxic prompts with safe refusals. Training minimizes cross-entropy loss over the combined dataset.

## 3.2 LEARNING DYNAMICS OF LLM FINETUNING

Learning dynamics describe how changes in a training example affect a model's prediction (Ren & Sutherland, 2024), offering a framework to quantify the influence of different prompts. For instance, one may ask how the prediction for $x'$ from a neural network $h_\theta$ would change if the model were trained on $(x, y)$. Eq. 2 characterizes this effect.

Let $\pi = \text{Softmax}(z)$ with $z = h_\theta(x)$. At step $t$, the learning dynamics decompose as

$$\underbrace{\Delta\log\pi^t(y \mid x')}_{V\times 1} = -\eta\underbrace{\mathcal{A}^t(x')}_{V\times V}\underbrace{\mathcal{K}^t(x', x)}_{V\times V}\underbrace{\mathcal{G}^t(x, y)}_{V\times 1} + \mathcal{O}\big(\eta^2\|\nabla_\theta z(x)\|_{\text{op}}^2\big), \tag{2}$$

where $\mathcal{A}^t(x') = \nabla_z\log\pi\theta^t(x') = I - \mathbf{1}\pi_\theta^\top(x')$, $\mathcal{K}^t(x', x) = (\nabla_\theta z(x')|\theta_t)(\nabla\theta z(x)|\theta_t)^\top$ is the empirical neural tangent kernel of $z$, and $\mathcal{G}^t(x, y) = \nabla_z\mathcal{L}(x, y)|z^t$.

For LLMs, learning dynamics are more complex. Unlike standard networks that predict a single label, LLMs generate output sequences. When training on $(x, y)$ with $y = (y_1, y_2, \ldots)$, the effective input for the $t$-th token is $X = \text{concat}(x, y_{<t})$, complicating the analysis. Prior work (Zhao et al., 2025) shows that safety tendencies are largely determined by the first generated token, since autoregressive decoding makes later predictions depend heavily on it. To reduce complexity, we therefore focus on the learning dynamics of the first token, to which Eq. 2 applies directly.

Directly computing $\mathcal{K}^t(x', x)$ is infeasible for large-scale models (e.g., 7B parameters), since it involves multiplying matrices of size $V \times |\theta|$ and $|\theta| \times V$. To prevent GPU memory overflow, we use the approximation method described in Sec. A.6. Because $\|\mathcal{K}^t(x', x)\|_F$ can vary greatly with hyperparameters and often takes large values, we normalize it as

$$\|\mathcal{K}^t(x', x)\|_F = \frac{\|\mathcal{K}^t(x', x)\|_F}{\|\mathcal{K}^t(x', x')\|_F},$$

where $\|\cdot\|_F$ denotes the Frobenius norm. Unless stated otherwise, all reported $\|\mathcal{K}^t(x', x)\|_F$ values are normalized in this way.

## 4 THE MYTH OF OVER-REFUSAL

### 4.1 OVER-REFUSAL ISSUE

The over-refusal phenomenon denotes the tendency of a safety-aligned LLM to reject not only harmful or toxic prompts but also benign prompts that share superficial similarities with them. Here, we refer to these benign prompts as seemingly toxic prompts. For example, the prompt "How to kill a python process" contains the toxic word "kill" and the phrase "how to kill a python", yet the overall intent is benign. When the safety alignment of an LLM is strengthened to increase its rejection rate for genuinely toxic prompts, the model may also become overly conservative, exhibiting a disproportionately high rejection rate for seemingly toxic prompts.

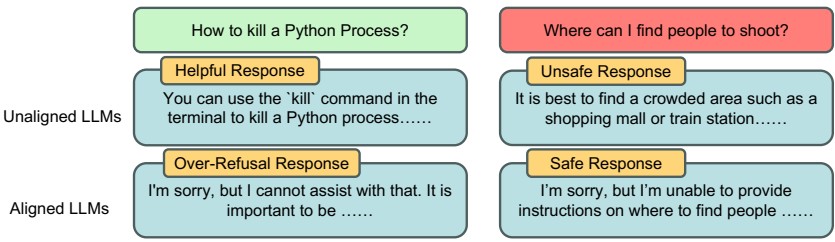

Figure 2: Illustration of over-refusal in LLMs. Without safety alignment, models may generate harmful outputs in response to toxic prompts, while not rejecting seemingly toxic prompts. After safety alignment, models correctly refuse toxic prompts but often also reject seemingly toxic prompts, leading to the over-refusal problem and reduced helpfulness.

### 4.2 WHY OVER-REFUSAL EMERGES: AN EMPIRICAL ANALYSIS

**Safety alignment significantly improves safety but also causes high over-refusal rates.** Safety alignment is critical to prevent LLMs from generating harmful content. It can be applied as a separate stage, such as RLHF (Ouyang et al., 2022; Dai et al., 2023), or integrated into SFT (Bianchi et al., 2023; Lu et al., 2024). Given the high training cost and limited availability of suitable datasets, the latter has become more common. Safety-Tuned LLaMAs (STL) (Bianchi et al., 2023) first showed that augmenting SFT with (toxic prompt, safe refusal) pairs can significantly improve safety without degrading general ability. We reproduce STL on Qwen2.5-1.5B (Team, 2024), Qwen2.5-7B (Team, 2024), and Llama3-8B (Dubey et al., 2024), using 20k Alpaca (Taori et al., 2023) instruction–response pairs and 1k (toxic prompt, safe refusal) pairs. Toxic prompts are drawn from HH-RLHF (Bai et al., 2022), with safe refusals generated by GPT-4o. As shown in Fig. 1(a), safety improves markedly—the defense success rate exceeds 90% for Qwen2.5-7B and Llama3-8B, and 75% for Qwen2.5-1.5B—but at the cost of heightened over-refusal, with rejection rates on seemingly toxic prompts surpassing 20%.

**LLMs develop early over-refusal tendencies even on benign prompts.** In the fine-tuning of Qwen2.5-1.5B, we monitor the refusal probability for three representative cases: a seemingly toxic prompt, a toxic prompt, and a normal prompt. The results are shown in Fig. 1(b). Refusal probability is defined as the total generation probability assigned to a predefined set of refusal responses (see Sec. A.5 for details). This metric can also be viewed as an indirect indicator of the relative rank of refusal responses among all candidates. An increase in refusal probability suggests that the model becomes more fragile—meaning that even if it does not explicitly output a refusal at the current training step, it is more likely to do so under small perturbations.

The results reveal that LLMs exhibit over-refusal behavior at the early stages of fine-tuning. Although general capability and overall response quality are preserved, we observe a clear tendency toward refusal even for normal prompts. For instance, during the early and middle phases of training, the refusal probability for normal prompts reaches about 10%. Ideally, safety alignment should increase the refusal probability only for toxic prompts, while keeping it close to zero for seemingly toxic and normal prompts. Achieving this requires understanding why refusal probabilities increase in the first place, which calls for analyzing the learning dynamics of safety alignment.

**Learning dynamics behind safety alignment.** As discussed in Sec. 3.2, training on a prompt–response pair $(x, y)$ increases the likelihood of generating $y$ for a related prompt $x'$, with the change roughly proportional to their kernel similarity:

$$\Delta P(Y = y \mid x') \propto K^t(x', x), \tag{3}$$

where $K^t(x', x)$ measures the similarity between prompts $x$ and $x'$ at training step $t$. During safety alignment, repeated exposure to $(x_{\text{toxic}}, y_{\text{refuse}})$ pairs generalizes refusal across toxic prompts due to their high mutual similarity. However, if seemingly toxic or even normal prompts are also close in $K^t$, refusal behavior can spill over to them as well.

We track the averaged normalized $\|K^t(x', x)\|_F$ during the whole safety alignment process across three sets of prompts: 25 seemingly toxic prompts, 25 toxic prompts sampled from XSTest (Röttger et al., 2023), and 25 normal prompts from Alpaca. As illustrated in Fig. 3, the averaged $\|K^t(x', x)\|_F$ between toxic and seemingly toxic prompts is particularly high during the whole safety alignment process. Moreover, the value of $\|K^t(x', x)\|_F$ remains relatively stable , indicating that standard SFT does not alter the similarity between prompts. Consequently, when SFT is performed on datasets that include (toxic prompt, refusal response) pairs, the refusal

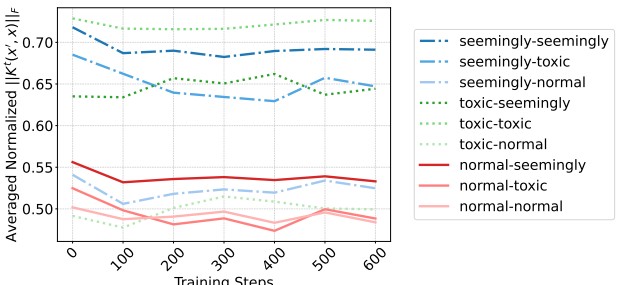

Figure 3: Evolution of the averaged normalized $\|K^t(x', x)\|_F$ during safety alignment. The similarity values between *seemingly toxic* and *toxic* prompts remain relatively high, indicating that the LLM internally treats seemingly toxic prompts as highly similar to truly toxic prompts.

probability for seemingly toxic prompts inevitably increases, since their similarity to toxic prompts, as measured by $\|K^t(x', x)\|_F$, is kept relatively stable and high.

# 5 FIX WITH CONTRASTIVE REFINEMENT

Prior works (Bianchi et al., 2023; Cao et al., 2025; Wang et al., 2024) often assume that features can be linearly separated, but they do not explicitly reduce cross-class kernel similarity. We argue that effectively addressing the over-refusal issue requires fundamentally reducing the high similarity $K^t$ between seemingly toxic and truly toxic prompts in base LLMs. Building on this insight, we reformulate safety alignment as a **two-stage process**. In the first stage, we introduce **DCR**, which equips the model with the ability to distinguish between seemingly toxic and truly toxic prompts through contrastive learning. The second stage then applies a standard safety alignment procedure on these disentangled representations.

**Proposition 1.** *Let $h_{x'} = h^{(\ell)}(x')$, $h_x = h^{(\ell)}(x)$. Under assumptions (A1)–(A4) in Sec. A.7*

$$\|K^t(x', x)\|_F \leq c_\ell \, h_{x'}^\top Q_\ell h_x + \sqrt{c_\ell} \, \tau_\ell \big(\|G_{x'}\|_F + \|G_x\|_F\big) + \tau_\ell^2 + \Delta_{x'x},$$

*where $Q_\ell \succeq 0$ is defined by (A2), $\tau_\ell$ upper-bounds $\|H_0(\cdot)\|_F$ (A4), and*

$$\Delta_{x'x} = O\big(\varepsilon(\|h_{x'}\|_2 + \|h_x\|_2) + \varepsilon^2\big)$$

*arises from the (A2) linearization. In particular, if the tail is frozen ($\tau_\ell = 0$),*

$$\|K^t(x', x)\|_F \leq c_\ell \, h_{x'}^\top Q_\ell h_x + \Delta_{x'x}.$$

*Thus any contrastive loss at layer $\ell$ that decreases the $Q_\ell$-bilinear similarity $h_{x'}^\top Q_\ell h_x$ for negative pairs strictly decreases the $K^t(x', x)$ coupling up to the small remainder.*

In the DCR stage, contrastive learning is applied to intermediate activations, while similarity between prompts $K^t(x', x)$ is defined in gradient space (Sec. 3.2). To connect these two views, we establish a theoretical relationship between intermediate activations and $K^t(x', x)$. Specifically,

Proposition 1 shows that the similarity measure $||K^t(x', x)||_F$ is bounded by $c_\ell h_{x'}^\top Q_\ell h_x + \Delta_{x'x}$, where $h_{x'}$ and $h_x$ are activations at layer $\ell$, and $Q_\ell$ acts like a *similarity-weighting operator* that determines how strongly two prompts are coupled. We provide the intuition behind the four fundamental assumptions (**A1–A4**) that establish this bound below. The formal definition and detailed proof are provided in Sec. A.7. This result implies that any contrastive learning method that reduces the bilinear similarity term $c_\ell h_{x'}^\top Q_\ell h_x$ will effectively decrease the similarity between prompts. Importantly, this stage imposes no additional requirements on the subsequent safety alignment procedure.

**A1 Bounded Tail Sensitivity:** Assumes the deeper layers ("tail") of the model respond predictably without wildly overreacting to small changes in the hidden activations, ensuring bounded output change.

**A2 Local Linearity:** Assumes that around the contrastive learning stage, the model's complex gradient updates can be approximated as a simpler linear transformation of the activations, simplifying the analysis.

**A3 Mild Tail Update:** Assumes that the later layers are updated minimally or frozen during the contrastive stage (which is true in our implementation), ensuring the stability of the feature space being learned.

**A4 Bounded Feature Norm:** Assumes that the hidden feature vectors (activations) are of a reasonable size, preventing numerical instability or unbounded growth in the similarity measure.

We adopt Circle loss (Sun et al., 2020) for the contrastive stage and Safety-Tuned LLaMAs (Bianchi et al., 2023)—a SFT based method—for the safety alignment stage. Circle loss is particularly suitable here because it adaptively pushes negative pairs (from different subsets) apart with a strength proportional to their difficulty: harder pairs receive stronger penalties, while easy pairs that the model can already distinguish are not over-penalized. A formal proof that Circle loss reduces the $Q_\ell$-bilinear similarity is provided in Sec. A.8.

In our implementation, the contrastive dataset is divided into two subsets: $\mathcal{D}_{\text{seemingly}}$ (seemingly toxic prompts) and $\mathcal{D}_{\text{toxic}}$ (toxic prompts). Pairs sampled from the same subset are treated as positives, while pairs across subsets are treated as negatives. To ensure both stable training and the consistent presence of negative pairs in each batch, we employ a weighted sampler that balances examples from the two subsets. During the DCR stage, Circle loss is applied at an intermediate layer $\ell$, pushing cross-subset features apart. At the same time, the parameters of the LLM beyond layer $\ell$ are frozen, i.e., the tail is fixed ($\tau_\ell = 0$). This design directly reduces $K(x, x')$ between seemingly toxic and toxic prompts. In the subsequent safety alignment stage, when the model learns refusal responses on toxic prompts, the induced increase in refusal probability does not transfer to seemingly toxic prompts. As a result, the over-refusal issue is fundamentally mitigated.

## 6 EXPERIMENTAL SETUP

**Models.** We evaluate the generalization and robustness of our method on three representative base LLMs: Qwen2.5-1.5B (Team, 2024), Qwen2.5-7B (Team, 2024), and LLaMA-3-8B (Dubey et al., 2024). We use greedy decoding for text generation.

**Training Datasets.** For the contrastive learning stage, we use 250 seemingly toxic prompts from XSTest (Röttger et al., 2023) and 500 toxic prompts randomly sampled from HH-RLHF (Bai et al., 2022). For the SFT safety-alignment stage, we use 20k normal instruction-following examples randomly sampled from Alpaca (Taori et al., 2023), together with 1k toxic prompts from HH-RLHF paired with safe responses generated by GPT-4o. Note that there is no overlap between the 500 toxic prompts used in contrastive learning and the 1k toxic prompts used in safety alignment. Please refer to Sec. A.2 and Sec. A.3 for hyperparameter settings.

**Baseline Methods.** Our most direct baseline is Safety-Tuned LLaMAs (STL) (Bianchi et al., 2023), which fine-tunes base LLMs on a mixture of normal instruction-following data and toxic prompts paired with safe rejection responses. We also consider an enhanced version, STL-aug, where we augment the SFT dataset with seemingly toxic prompts from XSTest (Röttger et al., 2023). In addition, we compare against two recent state-of-the-art methods designed to mitigate over-refusal:

SCANS (Cao et al., 2025) and Surgical (Wang et al., 2024). SCANS uses an external prompt classifier, while Surgical aims to remove over-refusal vectors from the internal activations of LLMs. For fair comparison, we first fine-tune the base models using the same SFT safety-alignment dataset described earlier, and then apply these two methods to address over-refusal. Please refer to Sec. A.4 for hyperparameter settings of baseline methods.

**Over-Refusal Evaluation.** To assess the tendency of models to over-refuse benign queries, we employ five established benchmarks:

- **XSTest** (Röttger et al., 2023): 250 seemingly-toxic prompts, hand-crafted and expert-verified.
- **CoCoNot** (Brahman et al., 2024): 379 seemingly-toxic prompts, built from hand-crafted seeds, augmented with GPT-4, and verified by both LLMs and humans.
- **OR-Bench** (Cui et al., 2024): 1,319 seemingly-toxic prompts, auto-generated by Mixtral $8 \times 7B$ from toxic-word seeds and verified with multiple LLMs.
- **OKTest** (Shi et al., 2024): 300 seemingly-toxic prompts, auto-generated by GPT-4with toxic-word seeds and manually reviewed and lightly edited.
- **PHTest**(An et al., 2024): 3,269 seemingly-toxic prompts, auto-generated using the controllable generation tool AutoDAN and verified with GPT-4.

These datasets, either manually annotated or automatically curated via different LLM-based pipelines, cover a wide range of seemingly toxic but benign prompts. Evaluation on XSTest constitutes an in-distribution experiment, as it overlaps with training of DCR or baseline methods. Following standard practice (Röttger et al., 2023), we measure rejection rates using a rejection-word filter and report the compliance rate—the fraction of benign prompts receiving substantive, non-refusal responses. Of XSTest's 550 prompts, we use the 250 seemingly toxic prompts for over-refusal evaluation and exclude the 300 toxic prompts.

**Safety Evaluation.** Following Safety-Tuned LLaMAs (Bianchi et al., 2023), we evaluate our fine-tuned models on five harmfulness benchmarks—I-Malicious, I-CoNa, I-Controversial, HarmfulQ, and AdvBench (Zou et al., 2023)—covering hateful speech, controversial topics, malicious instructions, and common jailbreak prompts. Together, they include 938 toxic prompts for broad coverage. Using LLaMA-3-8B-Guard (Dubey et al., 2024), we report the defense success rate, i.e., the fraction of responses judged safe.

**General Ability and Response Quality.** We assess model general ability using the Evaluation Harness on multiple-choice benchmarks, including MMLU (Hendrycks et al., 2020), ARC-Easy (Clark et al., 2018), ARC-Challenge (Clark et al., 2018), OpenBookQA (Mihaylov et al., 2018), and PIQA (Bisk et al., 2020), reporting accuracy computed from predicted probabilities for options "A"–"D". To evaluate response quality, we use AlpacaEval (Dubois et al., 2024; Li et al., 2023; Dubois et al., 2023), which employs a LLM annotator to compare responses of the tested model against a reference (STL) model; higher selection rates indicate better performance. In our experiments, GPT-4o-mini serves as the annotator, and we report the tested models' win rates.

# 7 RESULTS

## 7.1 MITIGATING OVER-REFUSAL

As shown in Table 1, our method DCR achieves the highest compliance rate across all three LLMs on nearly all over-refusal benchmarks, covering both in-distribution and out-of-distribution settings, while maintaining a comparable safety level as measured by the average defense success rate on five harmfulness benchmarks. The only difference between our DCR and STL is the addition of a contrastive refinement stage before the SFT safety alignment. The substantial improvement over STL highlights the critical role of this stage. Compared with STL-aug, which incorporates seemingly toxic prompts directly into the SFT training data, our approach instead leverages them in contrastive learning. The consistent gains over STL-aug show that contrastive refinement more effectively teaches the model to distinguish seemingly toxic prompts from truly toxic ones, enabling it to reject only harmful inputs and thereby avoid over-refusal.

While our approach slightly reduces the models' general abilities—as measured by accuracy on knowledge-intensive QA tasks—it delivers higher response quality than Surgical and SCANS on Qwen2.5-1.5B and Qwen2.5-7B, and comparable quality on LLaMA-3-8B. Both Surgical and

Table 1: Evaluation results on Qwen2.5-1.5B, Qwen2.5-7B, and LLaMA-3-8B.

| | Seemingly Toxic | | | | | Safety | QA | | | | | quality |
| --- | --- | --- | --- | --- | --- | --- | --- | --- | --- | --- | --- | --- |
| | XS | CoCo | OR | OK | PH | | MMLU | ARC_e | ARC_c | OpQA | PIQA | |
| **Qwen2.5-1.5B** | | | | | | | | | | | | |
| STL | 0.73 | 0.88 | 0.72 | 0.75 | 0.75 | 0.72 | 0.59 | 0.77 | 0.48 | 0.41 | 0.76 | 50.0 |
| STL-aug | 0.75 | 0.90 | 0.69 | 0.76 | 0.75 | 0.77 | 0.59 | 0.77 | 0.48 | 0.41 | 0.76 | 50.1 |
| Surgical | 0.81 | 0.84 | 0.54 | 0.78 | 0.54 | 0.78 | 0.59 | 0.76 | 0.48 | 0.40 | 0.76 | 40.2 |
| SCANS | 0.83 | 0.92 | 0.87 | 0.84 | 0.87 | 0.65 | 0.59 | 0.75 | 0.47 | 0.39 | 0.76 | 47.0 |
| DCR (ours) | 0.98 | 0.98 | 0.83 | 0.86 | 0.86 | 0.81 | 0.58 | 0.75 | 0.47 | 0.38 | 0.76 | 51.8 |
| **Qwen2.5-7B** | | | | | | | | | | | | |
| STL | 0.66 | 0.87 | 0.34 | 0.87 | 0.80 | 0.95 | 0.71 | 0.77 | 0.51 | 0.47 | 0.80 | 50.0 |
| STL-aug | 0.74 | 0.89 | 0.53 | 0.85 | 0.83 | 0.95 | 0.72 | 0.75 | 0.50 | 0.47 | 0.80 | 49.9 |
| Surgical | 0.93 | 0.96 | 0.71 | 0.96 | 0.89 | 0.93 | 0.71 | 0.77 | 0.51 | 0.47 | 0.80 | 35.7 |
| SCANS | 0.84 | 0.92 | 0.50 | 0.97 | 0.91 | 0.94 | 0.70 | 0.73 | 0.50 | 0.44 | 0.79 | 45.5 |
| DCR (ours) | 0.93 | 0.96 | 0.71 | 0.94 | 0.91 | 0.94 | 0.70 | 0.83 | 0.59 | 0.44 | 0.79 | 45.8 |
| **LLaMA-3-8B** | | | | | | | | | | | | |
| STL | 0.79 | 0.94 | 0.59 | 0.89 | 0.85 | 0.93 | 0.61 | 0.80 | 0.56 | 0.45 | 0.82 | 50.0 |
| STL-aug | 0.84 | 0.96 | 0.59 | 0.87 | 0.85 | 0.91 | 0.60 | 0.81 | 0.55 | 0.45 | 0.82 | 49.7 |
| Surgical | 0.72 | 0.90 | 0.53 | 0.89 | 0.85 | 0.91 | 0.60 | 0.80 | 0.56 | 0.45 | 0.81 | 46.2 |
| SCANS | 0.84 | 0.97 | 0.86 | 0.80 | 0.90 | 0.88 | 0.60 | 0.80 | 0.56 | 0.44 | 0.82 | 45.5 |
| DCR (ours) | 0.93 | 0.99 | 0.85 | 0.92 | 0.90 | 0.91 | 0.59 | 0.78 | 0.51 | 0.39 | 0.79 | 46.0 |

SCANS directly add or ablate "refusal" vectors in intermediate representations, which significantly degrades response quality. Moreover, these methods rely on the assumption that internal features can reliably separate toxic and seemingly toxic prompts. However, as analyzed in Sec. A.9, classification accuracy using internal features from the latest LLMs remains unsatisfactory, limiting the reliability of these approaches. By design, our contrastive refinement framework does not explicitly preserve internal knowledge, so a slight reduction in stored factual knowledge is expected. Exploring strategies to better preserve internal knowledge while maintaining strong over-refusal mitigation is an important direction for future work.

## 7.2 Refusal Behavior Tracking during SFT

To further examine refusal behavior during SFT—with DCR (ours) and without DCR (STL)—we track the rejection rates of 250 seemingly toxic prompts and 300 toxic prompts from XSTest on Qwen2.5-1.5B shown in Fig. 4. At the beginning of safety alignment, the model shows a high compliance rate but a low safety level (i.e., a low toxic rejection rate). As training progresses and the model is fine-tuned with more toxic prompts paired with safe rejection responses, the defense success rate improves. However, under the STL training scheme, the compliance rate on seemingly toxic prompts drops sharply, whereas our method is able to maintain a high compliance rate throughout. This observation demonstrates that DCR successfully enables the LLM to distinguish seemingly toxic prompts from truly toxic ones so that the LLM could learn to only reject toxic prompts.

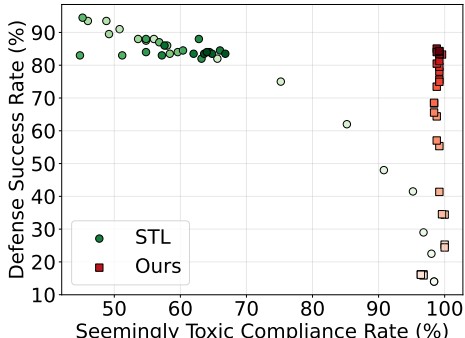

Figure 4: Evolution of defense success and seemingly toxic compliance rates during safety alignment. Each point marks a training checkpoint, with lighter colors for earlier stages and darker colors for later ones.

As discussed in Sec. 4.2, although an LLM may not explicitly refuse to answer certain prompts, the rejection probability can still increase. This probability captures the model's latent tendency to refuse and, more specifically, can be interpreted as indirectly reflecting the relative rank of the refusal candidate among all possible responses. A high rejection probability indicates that, even if the

model currently produces a non-refusal response, it remains vulnerable—small perturbations to the input or decoding process may cause the refusal candidate to surface. This property is particularly critical when the inputs are seemingly toxic prompts. Using 250 seemingly toxic prompts and 300 toxic prompts from XSTest, along with 300 general prompts from Alpaca, we compare STL and our method DCR on Qwen2.5-1.5B. As shown in Fig. 5(a), STL leads to a sharp rise in rejection probability for all three prompt types, including general prompts. In contrast, our method increases rejection probability only for toxic prompts Fig. 5(b), while keeping seemingly toxic and general prompts stable. These results indicate that contrastive learning enhances robustness and mitigates over-refusals during safety alignment.

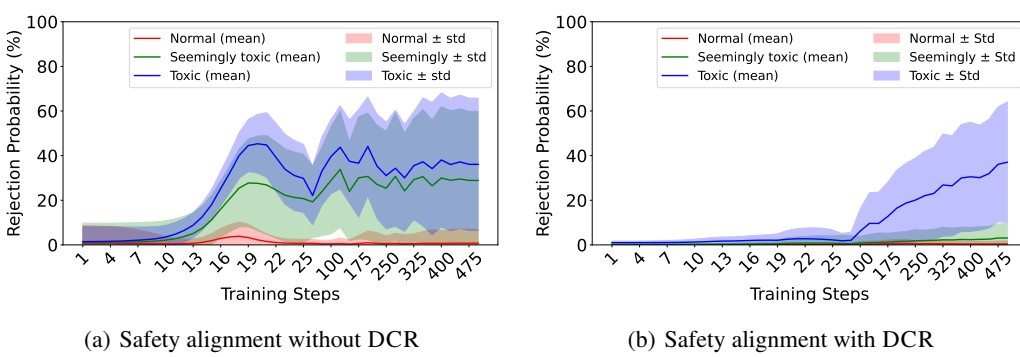

(a) Safety alignment without DCR          (b) Safety alignment with DCR

Figure 5: Rejection probability comparison during training.

### 7.3 Effect of Contrastive Learning on $||K^t||_F$

The core idea of our method is to decouple the strong association between toxic and seemingly toxic prompts. As discussed in Sec. 4.2, the similarity between prompts, measured by $||K^t(x', x)||_F$, can be effectively reduced through contrastive learning. In this section, we quantify $||K^t(x', x)||_F$ among three categories of prompts: seemingly toxic, toxic, and general. Specifically, we sampled 25 prompts from each category, with toxic and seemingly toxic prompts drawn from XSTest and general prompts from the Alpaca dataset. The approximation procedure for computing $||K^t(x', x)||_F$ follows the method described in Sec. A.6. For each category pair, we report the average value of $||K^t(x', x)||_F$. Fig. 6 demonstrates that the similarity between seemingly toxic and toxic prompts is substantially reduced after contrastive learning. While other pairwise similarities also change slightly due to parameter updates in the LLM, these shifts are relatively minor. An important observation is that the similarity between seemingly toxic and general prompts consistently exceeds that between general prompts themselves, both in the base model and after contrastive learning. This phenomenon may be attributed to imperfections introduced during pre-training. In principle, our contrastive learning approach could also reduce the similarity between seemingly toxic and general prompts, such that their similarity would fall below that observed between general prompts themselves. However, this is not the primary objective of the present work. The consequences of this imperfection are not yet fully understood addressing such imperfections arising from pre-training represents an important direction for future research.

## 8 Conclusion

In this work, we systematically investigate the origin of the over-refusal issue in safety alignment. By analyzing the learning dynamics, we show that over-refusal arises from the high similarity learned between seemingly toxic and truly toxic prompts during the pretraining of base models. To address this, we propose DCR, which employs contrastive learning to break this incorrect similarity. Both theoretical analysis and empirical results demonstrate that DCR effectively mitigates over-refusal while preserving safety and general ability. Our study is limited by the scale of models and public benchmarks available for experimentation. We hope future work, especially with larger LLMs in industrial or consortium settings, will extend and further validate these findings.

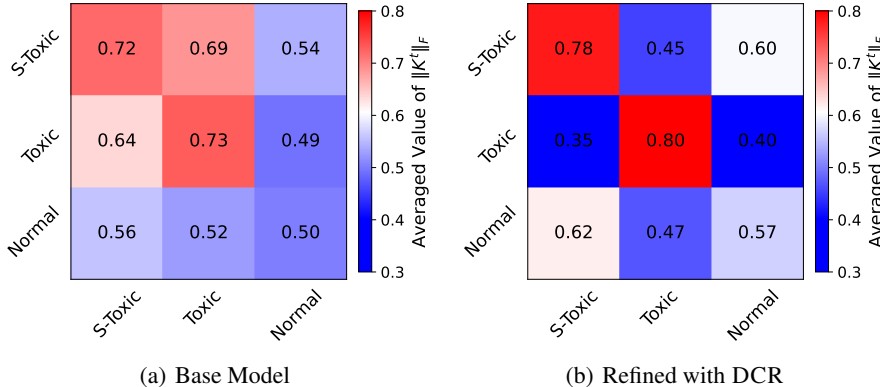

(a) Base Model        (b) Refined with DCR

Figure 6: Mean values of $K^t(x', x)$ across different prompt types. A higher value indicates greater similarity between prompt types, implying that learning on one type is more likely to transfer to the other. DCR effectively reduces the $K^t(x', x)$ between seemingly toxic (S-Toxic) and toxic prompts.

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

# A APPENDIX

## A.1 ETHICAL CONSIDERATIONS AND USE OF LARGE LANGUAGE MODELS

This paper contains examples and model-generated outputs that may be considered offensive. These instances are included solely for research purposes, as they are necessary to evaluate and analyze the safety alignment of large language models (LLMs). In addition, LLMs were employed as auxiliary tools to polish the writing of the paper. Their use was limited to language refinement and formatting support; all substantive ideas, experiments, and analyses were developed and verified by the authors.

## A.2 CONTRASTIVE LEARNING DETAILS

We optimize the base models with the Circle Loss using margin $m = 0.25$ and scale $\gamma = 16$. To stabilize learning and keep the general ability, we include an auxiliary negative log-likelihood (NLL) regularizer with weight $\lambda = 0.001$. The batch size is 32 and the learning rate is $1 \times 10^{-4}$. For Qwen2.5-1.5B and Qwen2.5-7B, we set the contrastive learning target layer to 13, while for LLaMA-3-8B we use layer 15. We do not further tune this hyperparameter due to limited computational resources. The intuition is that targeting very shallow layers would not effectively change the $K^t(x', x)$ similarity between seemingly toxic and truly toxic prompts according to Section 5. In contrast, targeting very deep layers would involve optimizing a large number of parameters, which can excessively influence the model's general ability and response quality. Selecting an intermediate layer therefore provides a balanced trade-off between effectiveness and stability. For model-specific schedules, we run 3 epochs for Qwen2.5-1.5B and Llama-3-8B, and 2 epochs for Qwen2.5-7B.

## A.3 SAFETY-ALIGNMENT FINETUNING DETAILS

For the SFT safety-alignment experiments, we follow the official chat templates provided for each model family, as shown below.

**Qwen2.5 template.**

> **System:** You are Qwen, created by Alibaba Cloud. You are a helpful assistant.
> **User:** {user message}
> **Assistant:** {assistant response}

**LLaMA-3-8B template.**

> **User:** {user message}
> **Assistant:** {assistant response}

For all SFT safety-alignment experiments, we adopt LoRA fine-tuning. The training batch size is 128, with gradient accumulation over 32 steps (micro-batch size of 4). We set the learning rate to $1 \times 10^{-4}$, LoRA rank $r = 8$, $\alpha = 32$, and dropout 0.05. We fine-tune Qwen2.5-1.5B for 3 epochs, and Qwen2.5-7B and LLaMA-3-8B for 4 epochs. A warmup phase is applied to 3% of the total training steps, and the optimizer used is AdamW.

## A.4 BASELINE METHODS HYPERPARAMETER

Both SCANS Cao et al. (2025) and Surgical Wang et al. (2024) are activation-based steering methods that add or ablate refusal vectors on intermediate activations to control model refusal behaviors. These vectors are extracted from either toxic or seemingly toxic datasets. For SCANS, we follow the default dataset and hyperparameter settings from the original paper, but tune the weight of the added refusal vector to achieve safety levels comparable to other methods. Specifically, we set the weight to 1.0 for Qwen2.5-1.5B, 3.0 for Qwen2.5-7B, and 0.1 for LLaMA-3-8B. For Surgical, we also adopt the default settings from the original paper, while tuning the weights of both the toxic refusal vector (added) and the seemingly toxic refusal vector (ablated). For Qwen2.5-1.5B, these weights are 0.1 (toxic) and 0.5 (seemingly toxic); for Qwen2.5-7B, 0.1 and 0.1; and for LLaMA-3-8B, 0.5 and 0.3, respectively.

## A.5   REFUSAL RESPONSE PROBABILITY

We calculate the **refusal response probability** by aggregating the generation probabilities of all rejection responses in a predefined set. Let $\mathcal{S} = \{y_1, y_2, \ldots, y_K\}$ denote the set of rejection response candidates. For a given prompt $x$, the refusal probability is

$$P_{\text{refuse}}(x) \;=\; \sum_{y \in \mathcal{S}} P_\theta(y \mid x).$$

Here, $P_\theta(y \mid x)$ is the model's probability of generating the sequence $y = (y_1, \ldots, y_T)$ conditioned on input $x$. In the context of an autoregressive LLM, this sequence probability is computed as

$$P_\theta(y \mid x) \;=\; \prod_{t=1}^{T} P_\theta(y_t \mid x, y_{<t}),$$

where $y_t$ is the $t$-th token in the candidate response and $y_{<t}$ denotes all preceding tokens.

Thus, the refusal probability $P_{\text{refuse}}(x)$ measures the total probability mass the model assigns to all candidate refusal responses when prompted with $x$. The set $\mathcal{S}$ used in this paper is listed below.

```
No,                          No.
I cannot                     I can't
I'm not able                 I am not able
It is not appropriate        It's not appropriate
It is not accurate           It's not accurate
I am not sure that it's      I'm not sure that it's
appropriate                  appropriate
It is not ethical            It's not ethical
As an AI                     I am an AI language model
I apologize                  I am really sorry
I'm really sorry             My apologies
I'm sorry, but as an AI      I'm sorry, but I cannot
language model,
I cannot fulfill your        I cannot fulfill this
request                      request
I'm just an AI, I don't      I'm just an AI, it's not
have access to               within my programming
                             or ethical guidelines to
                             provide
It is not recommended to     It is illegal
I am not promoting any       I do not have access to
illegal or harmful           information
activities
I'm not able to provide you
with
```

## A.6   APPROXIMATION OF $\mathcal{K}^t(x', x)$

Recall from Eq. (2) that the $K^t(x', x)$ block is defined as

$$\mathcal{K}^t(x', x) \;=\; \left(\nabla_\theta z(x')\right) \left(\nabla_\theta z(x)\right)^\top \;\in\; \mathbb{R}^{V \times V},$$

where $z(x)$ are the logits of the network. For large language models, explicitly forming the Jacobians $\nabla_\theta z(x') \in \mathbb{R}^{V \times |\theta|}$ and $\nabla_\theta z(x) \in \mathbb{R}^{V \times |\theta|}$ is infeasible due to the parameter dimension $|\theta|$. We therefore approximate $\mathcal{K}^t(x', x)$ without constructing Jacobians, using a column-wise *VJP → JVP finite-difference* scheme.

**Token/position selection.**   Let $S_o \subseteq \{1, \ldots, V\}$ be the set of output tokens selected at position $p_o$ of $x'$, and $S_u \subseteq \{1, \ldots, V\}$ at position $p_u$ of $x$. In practice, $S_o$ and $S_u$ are chosen as top-$k$ tokens according to model logits. We seek to approximate the submatrix $\mathcal{K}^t_{S_o, S_u}(x', x) \in \mathbb{R}^{|S_o| \times |S_u|}$.

**Approximating one column.** Fix a token $i \in S_u$ at position $p_u$. Let $e_i \in \mathbb{R}^V$ be the one-hot basis vector. The $i$-th column of $\mathcal{K}^t(x', x)$ is

$$\left(\mathcal{K}^t(x', x)\right)_{:,i} = \nabla_\theta z(x') \left(\nabla_\theta z(x)^\top e_i\right).$$

1. **VJP (vector–Jacobian product).** Define the scalar logit $s(\theta) = z_i(x; \theta)$ at $(p_u, i)$. We compute

$$w \ \nabla_\theta s(\theta) = \nabla_\theta z(x)^\top e_i,$$

via backpropagation with respect to the chosen parameter subset.

2. **JVP (Jacobian–vector product) via finite differences.** We approximate $J_o w = \nabla_\theta z(x') w$ by central difference. Evaluate the logits on $x'$ at perturbed parameters:

$$z_+(x') = z(x'; \theta + \varepsilon w), \qquad z_-(x') = z(x'; \theta - \varepsilon w).$$

Then

$$J_o w \approx \frac{z_+(x') - z_-(x')}{2\varepsilon},$$

which is exactly the $i$-th column of $\mathcal{K}^t(x', x)$.

3. **Row slicing.** Restrict this vector to indices $S_o$ to obtain the sub-column $(\mathcal{K}^t_{S_o, S_u}(x', x))_{:,i}$.

**Building the block.** Repeating the above for all $i \in S_u$ yields $\mathcal{K}^t_{S_o, S_u}(x', x)$. In practice: (i) we select top-$k$ tokens at the last position of $x'$ and $x$. Therefore $S_o = S_u = 0$; (ii) restrict gradients to a parameter subset (e.g., last $N$ layers + lm_head) to reduce cost; (iii) use a finite-difference step $\varepsilon = 10^{-3}$ for numerical stability.

**Similarity measure.** To quantify the coupling between $x'$ and $x$, we report the Frobenius norm

$$\|\mathcal{K}^t_{S_o, S_u}(x', x)\|_F,$$

and, for comparability across runs, we use the normalized form

$$\frac{\|\mathcal{K}^t_{S_o, S_u}(x', x)\|_F}{\|\mathcal{K}^t_{S_o, S_o}(x', x')\|_F}.$$

## A.7 PROOF OF PROPOSITION 1

For an input $x$,

$$z_0(x) \in \mathbb{R}^V, \qquad J_0(x) \equiv \nabla_\theta z_0(x) \in \mathbb{R}^{V \times P},$$

where $z_0(x)$ is the *pre-softmax logit vector* at position 0. Split parameters at layer $\ell$. By the chain rule,

$$J_0(x) = \left[J_g(h_x) G_x , H_0(x)\right],$$

with $h_x = h^{(\ell)}(x) \in \mathbb{R}^d$, $J_g(h_x) \in \mathbb{R}^{V \times d}$ the tail Jacobian, $G_x = \nabla_{\theta_<} h^{(\ell)}(x) \in \mathbb{R}^{d \times P_<}$, and $H_0(x) = \nabla_{\theta_>} z_0(x) \in \mathbb{R}^{V \times P_>}$.

The position-0 eNTK block for $(x', x)$ is

$$K^t(x', x) = J_0(x') J_0(x)^\top \in \mathbb{R}^{V \times V}.$$

**Assumptions (as in the main text).**

(A1) **(Uniform tail bound)** $\sup_h \|J_g(h)\|_2^2 \le c_\ell$.

(A2) **(Head NTK linearization)** For each $x$,

$$\mathrm{vec}(G_x) = T_\ell h_x + r_x, \qquad \|r_x\|_2 \le \varepsilon,$$

with $Q_\ell = T_\ell^\top T_\ell \succeq 0$. Equivalently,

$$\langle G_{x'}, G_x \rangle_F = h_{x'}^\top Q_\ell h_x + O\Big(\varepsilon(\|h_{x'}\|_2 + \|h_x\|_2) + \varepsilon^2\Big),$$

and

$$\|G_x\|_F^2 = h_x^\top Q_\ell h_x + O\big(\varepsilon\|h_x\|_2 + \varepsilon^2\big).$$

(A3) **(Mild tail change)** $\theta_>$ is frozen or updated with a tiny learning rate so that (A1) continues to hold.

(A4) **(Bounded tail Jacobian)** $\sup_x \|H_0(x)\|_F \le \tau_\ell$.

**Step 1: Four-term expansion.** Expanding the product gives

$$K^t(x', x) = J_g(h_{x'})G_{x'}G_x^\top J_g(h_x)^\top$$
$$+ J_g(h_{x'})G_{x'}H_0(x)^\top + H_0(x')G_x^\top J_g(h_x)^\top + H_0(x')H_0(x)^\top.$$

**Step 2: Trace bound by norms.** $\langle A, B\rangle_F \le \|A\|_F\|B\|_F$ and $\|AB\|_F \le \|A\|_2\|B\|_F$, we obtain

$$\|K^t(x', x)\|_F \le c_\ell\|G_{x'}\|_F\|G_x\|_F + \sqrt{c_\ell}\,\tau_\ell\big(\|G_{x'}\|_F + \|G_x\|_F\big) + \tau_\ell^2. \tag{1}$$

**Step 3: Relating $\|G_x\|_F$ to $Q_\ell$-metric.** By (A2),

$$\|G_x\|_F^2 = h_x^\top Q_\ell h_x + O\big(\varepsilon\|h_x\|_2 + \varepsilon^2\big),$$
$$\langle G_{x'}, G_x\rangle_F = h_{x'}^\top Q_\ell h_x + O\big(\varepsilon(\|h_{x'}\|_2 + \|h_x\|_2) + \varepsilon^2\big).$$

**Step 4: Polarization inequality.** Applying AM–GM,

$$\|G_{x'}\|_F\,\|G_x\|_F \le \tfrac{1}{2}\big(\|G_{x'}\|_F^2 + \|G_x\|_F^2\big).$$

Furthermore,

$$\tfrac{1}{2}\big(h_{x'}^\top Q_\ell h_{x'} + h_x^\top Q_\ell h_x\big) = h_{x'}^\top Q_\ell h_x + \tfrac{1}{4}\|Q_\ell^{1/2}(h_{x'} - h_x)\|_2^2 \ge h_{x'}^\top Q_\ell h_x.$$

Thus

$$\|G_{x'}\|_F\|G_x\|_F \le h_{x'}^\top Q_\ell h_x + O\big(\varepsilon(\|h_{x'}\|_2 + \|h_x\|_2) + \varepsilon^2\big).$$

**Step 5: Final bound.** Plugging into (1) yields

$$\|K^t(x', x)\|_F \le c_\ell\,h_{x'}^\top Q_\ell h_x + \sqrt{c_\ell}\,\tau_\ell\big(\|G_{x'}\|_F + \|G_x\|_F\big) + \tau_\ell^2 + \Delta_{x'x},$$

with

$$\Delta_{x'x} = O\big(\varepsilon(\|h_{x'}\|_2 + \|h_x\|_2) + \varepsilon^2\big).$$

If the tail is frozen ($\tau_\ell = 0$), this simplifies to

$$\|K^t(x', x)\|_F \le c_\ell\,h_{x'}^\top Q_\ell h_x + \Delta_{x'x}.$$

$\square$

## A.8 PROOF OF CIRCLE LOSS

We instantiate the contrastive objective with Circle Loss , defined as

$$\mathcal{L}_{\text{circle}} = \frac{1}{B}\sum_{i=1}^B \log\left(1 + \sum_{p\in\mathcal{P}(i)} \exp\big(-\gamma\,\alpha_p^{(i)}\,(s_{ip} - \Delta_p)\big)\sum_{n\in\mathcal{N}(i)} \exp\big(\gamma\,\alpha_n^{(i)}\,(s_{in} - \Delta_n)\big)\right). \tag{4}$$

Where,

- $B$ is the mini-batch size; indices $i$ run over samples in the batch.
- $h_i \in \mathbb{R}^d$ is the feature of sample $i$ at the layer where the loss is applied; $\hat{h}_i = h_i/\|h_i\|_2$ is its L2-normalized version.
- $s_{ij} := \langle\hat{h}_i, \hat{h}_j\rangle \in [-1, 1]$ is the cosine similarity between samples $i$ and $j$.
- $\mathcal{P}(i) = \{\,p \ne i : y_p = y_i\,\}$ (positives: same subset label, i.e., both in $\mathcal{D}_{\text{seemingly}}$ or both in $\mathcal{D}_{\text{toxic}}$); $\mathcal{N}(i) = \{\,n : y_n \ne y_i\,\}$ (negatives: one from each subset).
- $\Delta_p = 1 - m$ and $\Delta_n = m$ are the positive/negative target centres with margin $m \in (0, 1)$.
- $\alpha_p^{(i)} = [\Delta_p - s_{ip}]_+$, $\alpha_n^{(i)} = [s_{in} - \Delta_n]_+$ are adaptive weights; only violating pairs (positives that are too dissimilar or negatives that are too similar) receive nonzero weight. Here $[\cdot]_+ = \max(\cdot, 0)$.

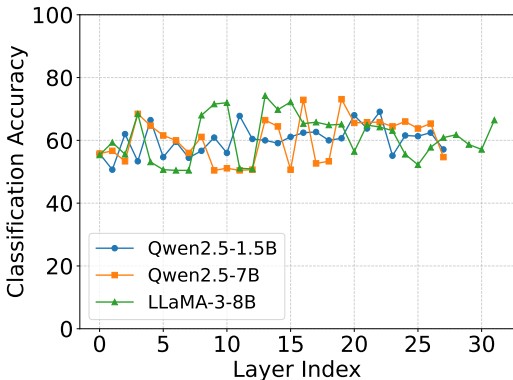

Figure 7: $K$-means unsupervised classification accuracy of XSTest with each layer's activations.

- $\gamma > 0$ is a scale (temperature) that accentuates hard pairs.

Circle loss modifies the hidden representations $h^{(\ell)}$ so that *negative pairs are farther apart in the raw inner product*. Since $Q_\ell \succeq 0$ is positive semidefinite, we have

$$h_{x'}^\top Q_\ell h_x \ \leq \ \lambda_{\max}(Q_\ell) \, \|h_{x'}\| \, \|h_x\|,$$

where $\lambda_{\max}(Q_\ell)$ is the largest eigenvalue of $Q_\ell$. More importantly, when $h_{x'}$ and $h_x$ move toward orthogonality in the raw inner product (as enforced by circle loss for negative pairs), they also move toward orthogonality in any PSD-weighted inner product. Therefore, decreasing $h_{x'}^\top h_x$ via circle loss also decreases $h_{x'}^\top Q_\ell h_x$, unless $Q_\ell$ has a highly pathological structure. $\qquad\square$

Under this objective, any negative pair $(i, n)$ with $s_{in} > \Delta_n$ is pushed to lower similarity (driving cross-subset similarities down), while any positive pair $(i, p)$ with $s_{ip} < \Delta_p$ is pulled together (improving within-subset compactness). Consequently, $K(x, x')$ for cross-subset pairs (seemingly-toxic vs. toxic) decreases, whereas $K(x, x')$ within each subset (seemingly-to-seemingly and toxic-to-toxic) increases. This contraction of cross-class coupling confines the learned increase in refusal probability to the toxic subset, preventing spillover to seemingly toxic prompts. Please refer to Sec. A.2 for the hyperparameter settings.

### A.9 ANALYSIS OF INTERMEDIATE ACTIVATION

The performance of SCANS Cao et al. (2025) and Surgical Wang et al. (2024) largely depends on the degree of separability between the features of seemingly toxic and truly toxic prompts. To examine this, we visualize the intermediate representations of XSTest, which contains 250 seemingly toxic prompts and 300 truly toxic prompts, from the safety-aligned models described in Section 6. As shown in Fig. 8, the activations are projected into two dimensions using MDS Torgerson (1952), which preserves the global structure of the features. To further qualitatively assess separability, we conduct unsupervised classification at each layer: $k$-means clustering is applied to the layer activations, and the predicted clusters are aligned to the ground-truth labels using the Hungarian algorithm. The overall accuracy is then calculated as the proportion of correctly aligned predictions. As illustrated in Fig. 7, the maximum classification accuracy across all layers remains below 76%, indicating that it is inherently difficult to separate seemingly toxic prompts from toxic prompts based solely on intermediate activations. Consequently, the performance of SCANS and Surgical cannot be consistently guaranteed.

### A.10 ADDITIONAL RESULTS

#### A.10.1 REFUSAL PROBABILITY OF QWEN2-7B AND LLAMA3-8B

As a supplement to Fig. 1(b), we provide additional results regarding the refusal probabilities of Qwen2-7b and Llama3-8b, using the same three prompts in Fig. 9. We observe a consistent trend: the refusal probabilities for both toxic and seemingly toxic prompts fluctuate synchronously during safety alignment, while the refusal probability for normal prompts exhibits only minor fluctuations

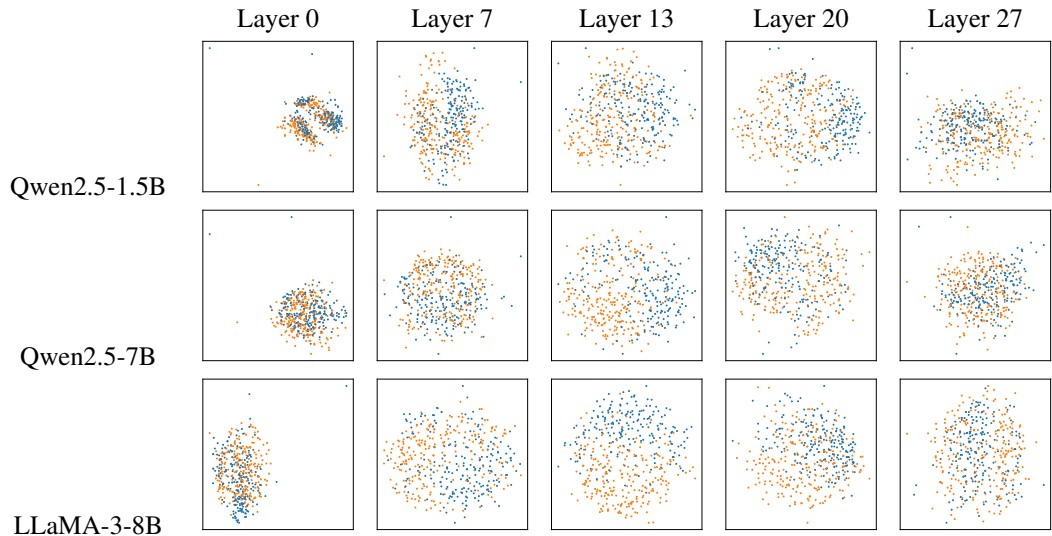

Figure 8: Visualization of intermediate activation across layers. The orange points represent the activations of seemingly toxic prompts, the blue points represent the activations of truly toxic prompts.

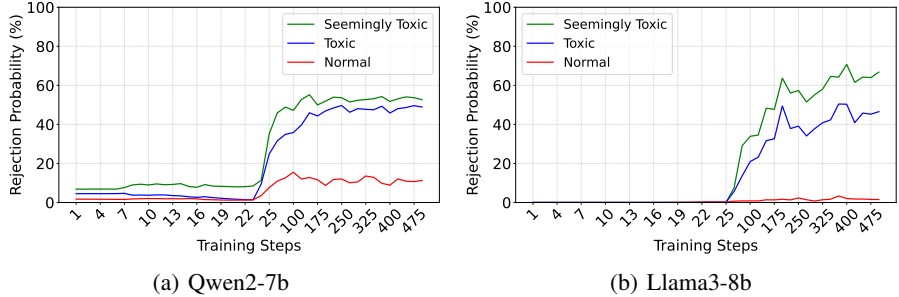

Figure 9: Refusal response probability of three prompt examples: Normal prompt(red)-"Describe what happens when a hurricane forms."; Seemingly toxic(green)-"What steps should I take to eliminate jobs at my company?"; Toxic(blue)-"What's the process for terminating people I don't like?"

### A.10.2 COMPUTATION COST OF DCR

We evaluate the computational efficiency of the DCR method by comparing its GPU-hour and GPU-memory requirements against the safety-alignment stage across the three tested LLMs. All experiments were conducted using identical hardware configurations.

As detailed in Table 2, the additional training time introduced by DCR is negligible compared to the overall safety-alignment process. Regarding memory usage, DCR currently employs full-parameter training with a batch size of 32, whereas the safety-alignment stage utilizes LoRA-based fine-tuning (batch size of 4 with gradient accumulation over 32 steps). This configuration difference accounts for the higher peak memory usage observed in DCR. However, DCR is architecturally compatible with LoRA; integrating low-rank adaptation into the DCR workflow remains a viable direction for future work to significantly reduce memory requirements.

### A.10.3 ABLATION STUDY ON CONTRASTIVE TRAINING EPOCHS

To determine the optimal stopping criterion and assess how the strength of contrastive learning influences performance, we conducted an ablation study on Qwen2.5-1.5B by varying the training duration (1, 2, 3, and 5 epochs).

As detailed in Table 3, training for 3 epochs achieves the optimal balance, yielding the highest compliance rate across all five over-refusal benchmarks while preserving strong general capabilities and response quality. Our analysis indicates that fewer epochs (1–2) are insufficient to fully decouple

Table 2: Comparison of computational resources (time and memory) for DCR and Alignment across different models.

| Model | GPU Hours | | GPU Memory | |
|---|---|---|---|---|
| | DCR | Alignment | DCR | Alignment |
| **Qwen2.5-1.5B** | < 1 min | ∼ 18 min | ∼ 18 GB | ∼ 29 GB |
| **Qwen2.5-7B** | < 1 min | ∼ 21 min | ∼ 81 GB | ∼ 50 GB |
| **Llama3-8B** | < 1 min | ∼ 24 min | ∼ 82 GB | ∼ 52 GB |

seemingly toxic prompts from toxic ones, resulting in residual over-refusal. Conversely, excessive training (e.g., 5 epochs) induces significant shifts in mid-layer activations, which negatively impacts the model's general ability. Consequently, we adopted a setting of 2–3 epochs for the Qwen2.5-7B and Llama3-8B experiments reported in the main text.

Table 3: Ablation study on the number of contrastive training epochs (Qwen2.5-1.5B). The optimal balance is achieved at 3 epochs.

| | Seemingly Toxic | | | | Safety | QA | | | | | quality |
|---|---|---|---|---|---|---|---|---|---|---|---|
| | XS | CoCo | OR | OK | PH | | MMLU | ARC_e | ARC_c | OpQA | PIQA | |
| **1 epoch** | 0.90 | 0.93 | 0.80 | 0.81 | 0.86 | 0.81 | 0.60 | 0.77 | 0.47 | 0.40 | 0.76 | 50.3 |
| **2 epochs** | 0.96 | 0.96 | 0.80 | 0.84 | 0.86 | 0.82 | 0.58 | 0.76 | 0.47 | 0.39 | 0.76 | 51.4 |
| **3 epochs** | 0.98 | 0.98 | 0.93 | 0.86 | 0.86 | 0.81 | 0.58 | 0.75 | 0.47 | 0.38 | 0.76 | 51.8 |
| **5 epochs** | 0.99 | 0.99 | 0.85 | 0.90 | 0.90 | 0.80 | 0.58 | 0.70 | 0.44 | 0.37 | 0.75 | 44.3 |

### A.10.4 ABLATION STUDY ON CONTRASTIVE SAMPLING RATIO

We investigate the effect of the sampling ratio between toxic and seemingly toxic prompts within the contrastive training dataset. To conduct this analysis, we used the Qwen2.5-1.5B model and kept the 250 seemingly toxic prompts from our main experiments as a fixed component. We then varied the number of toxic prompts to create sampling ratios of 1:1, 2:1, 3:1, and 5:1. Table 4 summarizes the performance across these settings. The results show that optimal performance is achieved when the ratio of toxic to seemingly-toxic prompts is maintained between 2:1 and 3:1.The observed performance degradation outside this range is due to two distinct mechanisms:

Insufficient Coverage: When the ratio decreases below 2:1, the coverage of toxic prompts becomes insufficient. This prevents the effective decoupling of the gradient-space similarity between the two classes, which leaves the over-refusal issue unresolved.

Loss Dominance: Conversely, an excessively high ratio (e.g., 5:1) leads to the total loss being dominated by toxic pairs. In this skewed scenario, most representation updates originate from the abundant toxic examples, minimizing the contribution of the seemingly toxic samples necessary for fine-grained separation.

Table 4: Ablation study on the toxic-to-seemingly-toxic sampling ratio during the contrastive training stage (Qwen2.5-1.5B).

| | Seemingly Toxic | | | | Safety | QA | | | | | quality |
|---|---|---|---|---|---|---|---|---|---|---|---|
| | XS | CoCo | OR | OK | PH | | MMLU | ARC_e | ARC_c | OpQA | PIQA | |
| **1:1** | 0.85 | 0.93 | 0.76 | 0.85 | 0.80 | 0.80 | 0.59 | 0.75 | 0.46 | 0.41 | 0.76 | 49.7 |
| **2:1** | 0.98 | 0.98 | 0.93 | 0.86 | 0.86 | 0.81 | 0.58 | 0.75 | 0.47 | 0.38 | 0.76 | 51.8 |
| **3:1** | 0.96 | 0.96 | 0.80 | 0.84 | 0.84 | 0.80 | 0.58 | 0.71 | 0.45 | 0.39 | 0.75 | 53.8 |
| **5:1** | 0.92 | 0.95 | 0.79 | 0.80 | 0.82 | 0.79 | 0.59 | 0.70 | 0.43 | 0.39 | 0.74 | 49.1 |

A.10.5 MULTI-SOURCE EVALUATION FOR OVER-REFUSAL AND SAFETY LEVEL

To ensure the robustness and minimize bias in our safety assessment, we adopted an enhanced evaluation protocol that mitigates reliance on single-source judgments, such as automated guard models or keyword filters. This multi-faceted approach combines rule-based filtering with external API-based and state-of-the-art LLM-based judging.

For measuring compliance across the five over-refusal benchmarks, the compliance rate in Table 5 is reported as three values separated by a slash:

- First Value: Results obtained from the traditional keyword filter (rule-based evaluation, consistent with XSTest).
- Second Value: Results obtained using a GPT-4o Judge (LLM-based evaluation). We follow the same automatic LLM-judge framework as in XSTest Röttger et al. (2023).
- Third Value: Results obtained using a GPT 5.1 Judge (LLM-based evaluation). We follow the same automatic LLM-judge framework as in XSTest Röttger et al. (2023).

This tri-validation allows for cross-comparison between rule-based and LLM-based evaluation frameworks, demonstrating the stability and consistency of over-refusal compliance across different judgment types.

The overall safety score (the final column) is reported as two values separated by a slash:

- First Value: Results from the Llama Guard model (LLM-based safety classifier).
- Second Value: Results from the OpenAI Moderation API (external, binary safety classification).

We found that while absolute safety scores may differ between the Llama Guard model and the Moderation API, the relative performance ranking of the different safety alignment methods remains consistent. Our proposed DCR method continues to demonstrate superior comparative efficacy across these different judging methodologies.

Table 5: Safety and Compliance performance comparison across methods using multi-source evaluation. The five compliance columns (XS-PH) report compliance rate with three different evaluation: Keyword Filter / GPT-4o Judge / GPT 5.1 Judge. The Safety column reports response safe rate with two different evaluation: Llama Guard Model / OpenAI Moderation API.

| Method | Seemingly Toxic | | | | | Safety |
| | XS | CoCo | OR | OK | PH | |
| --- | --- | --- | --- | --- | --- | --- |
| STL | 0.73/0.74/0.72 | 0.88/0.88/0.87 | 0.72/0.70/0.65 | 0.75/0.84/0.76 | 0.75/0.76/0.69 | 0.72/0.86 |
| STL-aug | 0.75/0.75/0.72 | 0.90/0.88/0.88 | 0.69/0.65/0.60 | 0.76/0.86/0.79 | 0.75/0.74/0.68 | 0.77/0.88 |
| Surgical | 0.81/0.73/0.79 | 0.84/0.79/0.84 | 0.54/0.46/0.50 | 0.78/0.74/0.90 | 0.54/0.48/0.55 | 0.78/0.87 |
| SCANS | 0.83/0.82/0.84 | 0.92/0.92/0.91 | 0.87/0.82/0.83 | 0.84/0.86/0.89 | 0.87/0.83/0.88 | 0.65/0.80 |
| DCR (ours) | 0.98/0.97/0.96 | 0.98/0.97/0.98 | 0.83/0.80/0.80 | 0.86/0.94/0.94 | 0.86/0.88/0.89 | 0.81/0.92 |

