# OpenReview forum: "Discern Truth from Falsehood: Reducing Over-Refusal via Contrastive Refinement"
_ICLR.cc/2026/Conference — ICLR 2026 Poster_

### Official Review · Reviewer_9fdp · 2025-10-21

**Soundness:** 3
**Presentation:** 3
**Contribution:** 2
**Rating:** 6
**Confidence:** 3

**Summary:**

The paper studies over‑refusal in safety‑aligned LLMs and proposes a two‑stage procedure, DCR (Discernment via Contrastive Refinement). Stage 1 applies a contrastive loss on intermediate representations to push *seemingly toxic* prompts away from *truly toxic* prompts while freezing later layers; Stage 2 runs standard SFT safety alignment. The paper argues that over‑refusal arises because these two prompt types are highly coupled; it provides a bound linking gradient‑space similarity ( |K_t(x',x)|\_F ) to a bilinear form on hidden states and shows empirical reductions of this coupling after DCR. Across three base models, DCR improves compliance on five over‑refusal test sets while keeping safety roughly unchanged. Figures 1, 3, 5–6 and Table 1 support the empirical story, and Appendices A, A–A detail implementation and metrics.

**Strengths:**

* **The method is clear and reproducible**. The two‑stage pipeline is easy to implement: Circle loss at an intermediate layer with the tail frozen, followed by standard SFT.
* **The evaluation has good coverage**. Results cover five over‑refusal benchmarks and five harmfulness suites, plus general‑ability tasks like AlpacaEval.
* **Theory-driven analysis**. Proposition 1 and Appendix A connect representation‑level contrastive changes to a decrease in the gradient‑space kernel. The paper also tracks “rejection probability” during training and shows that DCR raises it mainly for toxic prompts while keeping it stable for seemingly toxic prompts.

**Weaknesses:**

* **Framing of similarity as “incorrect.” (Line: 481)** The core motivation of this paper is that the similarity between seemingly toxic and toxic prompts is *incorrect* and should be decoupled. These prompts are expected to be similar at the representation level, and the issue is spillover during alignment. Therefore I am not very convinced about the motivation.
* **Pair construction and dataset confounding.** Seemingly toxic prompts come from XSTest while toxic prompts come from HH‑RLHF in the contrastive stage. This risks learning dataset or domain differences rather than toxicity level. An ablation that draws both classes from the same source or uses matched minimal pairs would directly test this concern.
* **Pairing strategy is weakly specified.** This is similar to the last point. Contrastive learning benefits from carefully curated *hard* negatives and matched positives. The paper samples pairs across subsets with a weighted sampler but does not align pairs by lexical template or topic, so the signal may be coarse. More rigorous pair‑wise design could strengthen the claim that DCR learns toxicity‑level separation rather than other differences.
* **Measurement choices may be brittle.** "Refusal probability" aggregates probability mass over a fixed list of refusal strings (Appendix A). This can miss paraphrases. It will be better to validate the refusal measurements with human annotators or LLM-judges.

Presentation suggestion: in Table 1 it's better to bold the best scores and clearly mark for each metric whether higher is better, so readers can read the trade‑offs at a glance.

**Questions:**

- How much of the gain remains when controlling for dataset origin? As far as I know, the seemingly toxic prompts from XS-Test and Over-Refusal usually have different sentence structures compared to those in toxic datasets.
- How do DCR’s cost and scalability compare to the baselines in Table 1?
- Could you provide an analysis of the reliability of the refusal response templates?
- I see that all baselines perform poorly on XS-Test. Is it possible that they are undertrained? Could you provide the results of the original model and a model overtrained using the baseline methods, so we can determine whether DCR is Pareto-optimal compared to the baselines?
- The first stage of DCR effectively penalizes high similarity. Does the normalized kernel similarity actually decrease after applying DCR? Figure 3 shows the similarity changes, but it appears to reflect a normal safety-alignment process rather than DCR specifically.

---

> ### Author Response · Authors · 2025-11-21
> **Part I**
>
> We sincerely thank the reviewer for their valuable comments and insights. Below, we address the points raised in the review. The additional experiments with larger-scale LLMs are currently in progress and will be posted in a subsequent update once completed.
> ## Core motivation for decoupling toxic vs. seemingly toxic similarity.
> Our argument is that—**from a safety perspective**—*seemingly toxic* prompts **should not share strong representational similarity** with *toxic* prompts. According to the learning dynamics of large language models (as shown in Equation 2), similar inputs tend to produce similar outputs during training. The similarity $K^t$ , defined in the gradient space (also known as the empirical Neural Tangent Kernel, or eNTK), captures this coupling.
>
> Our central motivation is that **during pretraining, LLMs learn overly high \( K^t \) similarity between toxic and seemingly toxic prompts**, which is theoretically incorrect for safety alignment. Due to the model’s generalization behavior, when LLMs are fine-tuned to refuse toxic prompts, they inevitably extend this refusal to seemingly toxic ones if the \( K^t \) similarity remains unbroken. This reasoning holds regardless of dataset composition, loss design, or alignment strategy, underscoring the necessity of **decoupling these representations** as proposed in our work.
>
> ## Measurement of refusal probability choices may be brittle.
>
> The list of refusal strings we used originates from **XSTest**, which provides a standardized and widely adopted string-matching protocol for detecting refusal responses. We further expanded this list by manually summarizing additional refusal patterns frequently observed during our experiments.
>
> There are two uses of this list:
> 1. **Refusal probability**, which measures the model’s *tendency* to refuse a given prompt, even if the final response is not explicitly a refusal. This metric reflects how likely a model is to move toward refusal in its response space.
> 2. **Refusal rate**, which measures the actual proportion of responses that contain explicit refusals within a benchmark.
>
> For the **refusal rate**, in addition to the string-matching method, we also employed **multi-LLM judgment** using *GPT-4o* and *GPT-5.1*, following the XSTest evaluation pipeline (see supplementary experiment in response to Reviewer 7tZR). The results from string matching and LLM-judgment are numerically consistent, supporting the **reliability** of our refusal prefix list.
>
> For the **refusal probability**, it is practically infeasible to compute exact probabilities since doing so would require labeling all possible responses as refusal or non-refusal. Even if we limited this to the top-100 responses, the computational cost would remain prohibitive. Therefore, as described in Appendix A.5, we approximate refusal probability by summing the generation probabilities of the fixed refusal prefixes. Expanding this list for broader linguistic coverage would improve accuracy, and we view this as a **community-level effort** for future work.
> ## Gains after controlling for dataset origin.
> We acknowledge that differences in sentence structure across datasets can influence learning efficiency and model performance. To examine this effect, we conducted an additional experiment on Qwen2.5-1.5B, varying the source of toxic prompts. In the main paper (Table 1), the seemingly toxic prompts for contrastive learning were drawn from XSTest, while the toxic prompts were taken from HH-RLHF. We introduced a new setting where both types of prompts were sourced from XSTest, resulting in two configurations: **“XSTest + HH-RLHF”** and **“XSTest both.”**
>
> As shown in the table below, the general ability and response quality remain comparable between the two settings. However, the **“XSTest both”** configuration exhibits a stronger over-refusal tendency, with lower compliance rates across the five over-refusal benchmarks. We attribute this to the **broader topical and syntactic coverage** of HH-RLHF, which enables **more effective decoupling** between toxic and seemingly toxic prompts, thereby mitigating over-refusal more thoroughly.
>
> | Method             | XS  | CoCo | OR  | OK  | PH  | Safety | MMLU | ARC_e | ARC_c | OpQA | PIQA | Quality |
> |--------------------|-----|------|-----|-----|-----|---------|------|-------|-------|------|------|----------|
> | XSTest both    | 0.88 | 0.96 | 0.76 | 0.82 | 0.85 | 0.80 | 0.59 | 0.76 | 0.47 | 0.41 | 0.76 | 51.1 |
> | XSTest + HHRLHF    | 0.98 | 0.98 | 0.93 | 0.86 | 0.86 | 0.81 | 0.58 | 0.75 | 0.47 | 0.38 | 0.76 | 51.8 |

---

> ### Author Response · Authors · 2025-11-21
> **Part II**
>
> ## Need better pair matching for contrastive learning.
> We agree that constructing more carefully matched positive and hard negative pairs can further enhance the effectiveness of contrastive learning. Sampling seemingly toxic and toxic prompts that share similar topics or overlapping lexical patterns would help the model focus more precisely on toxicity-related differences rather than unrelated features. We view this as an interesting and valuable direction for future work to improve the strategy of pair construction in contrastive training.
>
> ## DCR’s cost and scalability vs. baselines.
> As shown in the supplementary results (see our response to Reviewer 7tZR), the GPU-hours required by DCR are negligible (<1 min) compared to the safety-alignment stage (>20 min), while the GPU-memory requirement is relatively higher (~80GB v.s. ~50GB).  This difference arises because DCR is currently trained in a full-parameter setting with a batch size of 32 without gradient accumulation, whereas the safety-alignment stage uses LoRA-based fine-tuning with gradient accumulation over 32 steps and a batch size of 4. However, we note that DCR is fully compatible with LoRA, and integrating the two could further reduce GPU-memory consumption. Given time constraints and the paper’s primary focus on methodological contributions rather than efficiency optimization, we leave this LoRA-integrated efficiency extension as future work.
> ## Reliability of refusal templates.
> We follow the standard refusal response templates provided by XSTest, and further **expand this list** by manually summarizing additional refusal patterns frequently observed during our experiments. Including more templates would help achieve a more accurate measurement of refusal behavior. However, as the refusal template design is not the main contribution of this paper, we leave further refinement to future community efforts toward improving template coverage and diversity.
>
> To assess the **reliability** of our template-based string matching, we compared it against **multi-LLM judgment** using *GPT-4o* and *GPT-5.1*, following the XSTest automatic evaluation pipeline (see the supplementary experiment in response to Reviewer 7tZR). The results from string matching and LLM judgment are **numerically consistent**, supporting the robustness of our refusal prefix list. Due to limited time and resources, we were unable to conduct human evaluation, which we consider a valuable next step for future work.
>
> ## Possible undertraining of baselines.
> Among the four baselines used in our paper, two methods — Surgical and SCANS — are **training-free**. They directly modify prompt activations in intermediate layers during inference by adding or subtracting a *“refusal vector”* to control model behavior. However, the **root cause of over-refusal** lies in the model’s inability to distinguish seemingly toxic from toxic prompts. As a result, these methods may incorrectly apply the refusal vector to seemingly toxic inputs, amplifying over-refusal rather than mitigating it.
>
> For the **STL** baseline, similar patterns can be observed in prior work. In *Safety-Tuned LLaMAs* [1, Fig. 14] and *Semantic Loss Guided SFT* [2, Fig. 3], even 7B-scale models reject over 50% of seemingly toxic prompts in XSTest. Our findings are consistent with these results, indicating that **the poor performance is not due to undertraining**.
>
> For the **STL_aug** baseline, models are fine-tuned with safety-related datasets **augmented with XSTest** (see Section 6 of the main paper). We trained Qwen2.5-1.5B for 3 epochs, and Qwen2.5-7B and LLaMA3-8B for 4 epochs, whereas standard SFT usually runs for only 1–2 epochs. This ensures that the models are adequately trained and not underfit. However it's still not able to overfitted on the training set which include XSTest.
>
> Based on these results, we can conclude that the poor baseline performance on XSTest is **not due to insufficient training**, but reflects their inability to disentangle toxic and seemingly toxic representations — the core issue DCR is designed to address.
>
> **References:**
> [1] Bianchi F, Suzgun M, Attanasio G, et al. *Safety-tuned LLaMAs: Lessons from Improving the Safety of Large Language Models that Follow Instructions.* arXiv:2309.07875, 2023.
> [2] Lu Y, Sinha A, Varakantham P. *Semantic Loss Guided Data Efficient Supervised Fine Tuning for Safe Responses in LLMs.* arXiv:2412.06843, 2024.
> ## Whether DCR truly reduces kernel similarity.
>
> Please refer to **Figure 6** in **Section 7.3**, where we report the mean values of $K^t(x', x)$ across different prompt types. The results clearly demonstrate that the $K^t$ similarity between seemingly toxic and toxic prompts decreases substantially after applying contrastive learning**, confirming that **DCR effectively reduces representational coupling** in the middle layers.
>
> *We will update the PDF with our new results and typo fixes after performing all experiments asked by all reviewers.*

---

> > ### Comment · Reviewer_9fdp · 2025-11-25
> >
> > Thank you for your detailed response. While I still believe the motivation and method remain somewhat weak, the results make a meaningful contribution to the further study of seemingly toxic prompts. I will lean toward the opinions of the other reviewers.

---

> > > ### Author Response · Authors · 2025-11-26
> > >
> > > Thank you very much for your thoughtful follow-up and for taking the time to reassess our responses. While we understand and respect your remaining reservations regarding the motivation and methodological framing, we are genuinely grateful that you recognize the contribution our results make toward advancing the understanding of seemingly toxic prompts. We have revised the manuscript in accordance with the suggestions and discussions from all reviewers, and we remain committed to further improving the paper in the next revision cycle. Thank you again for your constructive engagement and for helping us strengthen the quality of this work.

---

### Official Review · Reviewer_w17s · 2025-10-29

**Soundness:** 3
**Presentation:** 3
**Contribution:** 3
**Rating:** 6
**Confidence:** 4

**Summary:**

This paper addresses the issue of over-refusal in LLM alignment. This paper focuses on the similarity between toxic prompts and seemingly toxic (but actually not toxic) prompts and shows that the rejection probability for these two types of prompts is highly correlated during the alignment process. To tackle the issue of over-refusal without losing the usefulness, the authors claim that it is important to distinguish truly toxic and seemingly toxic prompts and propose a two-stage alignment procedure: 1) contrastive learning between these prompts, then 2) standard safety alignment. Experiments show that contrastive learning effectively reduces the similarity between truly toxic and seemingly toxic prompts, and that it contributes to reducing the rate of refusal of seemingly toxic prompts while managing the refusal rate for truly toxic prompts.

**Strengths:**

It provides a novel insight into the over-refusal in LLM and provides a reasonable understanding of the cause of the over-refusal.

The proposed approach is well-motivated by the observation of high correlation between the refusal rates of truly toxic and seemingly toxic prompts.

The proposed approach is compared with several baseline approaches and the experiments show promising performance of the proposed approach.

**Weaknesses:**

The evaluation is done with relatively small models (up to 8B models) and the scalability is not shown.

I found the observations in Figures 1 and 3 are interesting and these are the cores of the proposed approach. However, this observation is provided only for a small (1.5B) model. To support the claim of this paper, it is recommended to add the observations for middle sized (7–8B) models.

The strength of the contrastive learning (such as the number of epoch, learning rate, etc.) may affect the performance significantly, which is not discussed in the main text.

**Questions:**

1. Do we observe similar trends as in Figures 1 and 3 with different models?

2. How much does the strength of the contrastive learning affect the performance? How can we decide when to finish? It looks nontrivial to me when to stop it.

3. Assumptions for Proposition 1 are not explained nor validated in the main text. Please provide the intuition and justification for them in the main text.

4. Lines in Figure 3 are hard to distinguish. Please use different line styles or so.

---

> ### Author Response · Authors · 2025-11-21
> **Part I**
>
> We sincerely thank the reviewer for their valuable comments and insights. Below, we address the points raised in the review. The additional experiments with larger-scale LLMs are currently in progress and will be posted in a subsequent update once completed.
>
> ## Whether similar patterns are observed in Figures 1 and 3 across different model architectures.
>
> For **Figure 1**, we provide additional experimental results on **Qwen2.5-7B** and **Llama3-8B** in **Figure 9** of the appendix. We record the refusal probabilities of the same three prompts used in Figure 1(b), and the same trend is observed: the refusal probabilities of *seemingly toxic* and *toxic* prompts **increase and decrease synchronously** during safety alignment, while the refusal probability of *normal* prompts also shows **minor fluctuations**. For **Figure 1(a)**, results from all three models are already included in the main text, so no revision is necessary.
>
> For **Figure 3**, it illustrates that the **\(K^t\) similarity** between different prompt types remains stable during SFT safety alignment, indicating that the LLM tends to perceive *seemingly toxic* prompts as similar to *toxic* ones. However, computing \(K^t\) defined in equation directly is **computationally intractable**, since it involves multiplying two gradient matrices of size \(|W| \times |V|\), where \(|W|\) is the number of model parameters and \(|V|\) is the vocabulary size. For instance, in **Qwen2.5-1.5B**, with \(|W| = 1.5\) billion and \(|V| = 151{,}643\), storing the full gradient matrix would require **over 900 terabytes of memory** (assuming 4 bytes per element).
>
> In **Figure 3**, we therefore compute the gradient only for the **last three layers** and retain the **top-64 tokens** in the vocabulary and use VJP-JVP trick to save memory (Please refer to appendix A.6), which already consumes about **130 GB**. Scaling this configuration to a **7B-parameter model** would require roughly
> $
> 130 \text{GB} \times \frac{7}{1.5} \approx 607 \text{GB},
> $
> making the computation impractical due to excessive storage and inter-GPU communication overhead. Consequently, it is **unfortunately not feasible** to reproduce Figure 3 for Qwen2.5-7B or Llama3-8B with current hardware constraints.
>
> ## How the strength of contrastive learning influences performance and how to determine an appropriate stopping criterion.
> To investigate how the strength of contrastive learning influences performance and how to determine an appropriate stopping criterion., we conducted an ablation study on **Qwen2.5-1.5B** by varying the number of training epochs for the contrastive stage (1, 2, 3, and 5 epochs).
>
> As shown in the table below, training for **3 epochs** achieves the best overall balance—yielding the **highest compliance rate** across the five over-refusal benchmarks while maintaining **strong general ability and response quality**. Fewer epochs (e.g., 1–2) are insufficient to fully decouple *seemingly toxic* and *toxic* prompts, leaving residual over-refusal issues. Conversely, excessive training (e.g., 5 epochs) overly shifts the mid-layer activations, which negatively impacts general ability and response quality.
>
> The results reported in **Table 1** for **Qwen2.5-7B** and **Llama3-8B** were also obtained using 2 or 3 epochs under this setting. Based on these observations, we conclude that **2–3 epochs** of contrastive training are sufficient and stable.
>
>
>
> | Epoch   | XS  | CoCo | OR  | OK  | PH  | Safety | MMLU | ARC_e | ARC_c | OpQA | PIQA | Quality |
> |----------|-----|------|-----|-----|-----|---------|------|-------|-------|------|------|----------|
> | **1 epoch** | 0.90 | 0.93 | 0.80 | 0.81 | 0.86 | 0.81 | 0.60 | 0.77 | 0.47 | 0.40 | 0.76 | 50.3 |
> | **2 epoch** | 0.96 | 0.96 | 0.80 | 0.84 | 0.86 | 0.82 | 0.58 | 0.76 | 0.47 | 0.39 | 0.76 | 51.4 |
> | **3 epoch** | 0.98 | 0.98 | 0.93 | 0.86 | 0.86 | 0.81 | 0.58 | 0.75 | 0.47 | 0.38 | 0.76 | 51.8 |
> | **5 epoch** | 0.99 | 0.99 | 0.85 | 0.90 | 0.90 | 0.80 | 0.58 | 0.70 | 0.44 | 0.37 | 0.75 | 44.3 |

---

> ### Author Response · Authors · 2025-11-21
> **Part II**
>
> ## The need to clarify and justify the assumptions underlying Proposition 1 in the main text.
> Here we provide the intuition behind the assumptions in Proposition 1 to aid understanding. Empirically validating these assumptions would be time-consuming and non-trivial, and such validation lies beyond the main scope of this paper. We leave it as an interesting direction for future work.
> 1. **Assumption A1 – Bounded Tail Sensitivity**
>    The deeper layers of the model do not overreact to small changes in the hidden activations; their output change is bounded.
>
> 2. **Assumption A2 – Local Linearity**
>    The gradients of the hidden representations can be well approximated by a linear transformation of the activations.
>
> 3. **Assumption A3 – Mild Tail Update**
>    During contrastive learning, the later (“tail”) layers are updated very little or kept frozen. It is true that we fix the deep layers paramter in our implementation.
>
> 4. **Assumption A4 – Bounded Feature Norm**
>    The hidden features (activations) have bounded norms across prompts.
>
> ## The visual distinction issue in Figure 3 and the suggestion to use different line styles for clarity.
>
> We thank the reviewer for the helpful suggestion regarding the visual clarity of Figure 3. We fully agree that using distinct line styles would make the figure easier to interpret. We plan to revise Figure 3 by adopting different line styles and markers to improve visual distinction once all additional experiments and responses to reviewers are completed. The updated version will be included in the final revision of the paper to ensure consistency across figures and analyses.
>
>
> *We will update the PDF with our new results and typo fixes after performing all experiments asked by all reviewers.*

---

> > ### Comment · Reviewer_w17s · 2025-11-25
> >
> > Thanks for the response. The responses are all reasonable. I will check the revised PDF once it is updated.
> >
> > Additional minor correction:
> > Line 232: "Alpaca.As illustrated in Fig. 3, " A space between Alpaca and As is missing.

---

> > > ### Author Response · Authors · 2025-11-26
> > >
> > > Thank you very much for your thoughtful feedback. We sincerely appreciate your acknowledgement of the improvements we have made to address your concerns. We are also grateful for your careful attention to formatting and typographical issues. Following your suggestions, we have thoroughly revised the manuscript to ensure clarity, correctness, and consistency throughout. We truly appreciate your engagement and support and if you have any additional suggestions or further questions, please feel free to let us know.

---

### Official Review · Reviewer_jZ3e · 2025-10-29

**Soundness:** 3
**Presentation:** 2
**Contribution:** 3
**Rating:** 8
**Confidence:** 3

**Summary:**

The authors tackle LLM overrefusal by contrastively reducing gradient coupling between toxic and seemingly toxic prompts during the pre-alignment training stage. They apply a contrastive loss at a single mid layer to decrease activation similarity. By their theoretical results, this method lowers the empirical neural tangent kernel similarity between these prompt types. Their downstream benchmarking results substantial improvements in compliance rates across multiple refusal benchmarks, while maintaining strong safety performance and capabilities on standard general knowledge tasks.

**Strengths:**

1. **Clear theoretical grounding.**
- The paper (Section 5, 7.3) provides a formal link between activation similarity and empirical neural tangent kernel similarity to show how their method lowers gradient coupling.

2. **The method is conceptually simple and doesn’t require architectural modification.** It targets directly the mechanism behind over-refusal (gradient coupling) rather than surface behaviors, which is underexplored.

**Weaknesses:**

1. **Rejection probability calculation can be biased.**
- As described in Appendix A.5, the refusal probability aggregates mass over a fixed list of refusal strings. Models can refuse with more nuanced paraphrases. Providing calibration such as precision/recall could help further support the robustness of this metric. Maybe adding a learned refusal classifier for this metric would be useful too.

2. **Formatting.**
- Citations that should be parenthetical are written as “Author, Year” instead of “(Author, Year)”. They should use `\citep` instead of `\citet`.
- In Table 1 on page 8, **bolding** best performance will make the results more clear.

3. **Computational costs.**
- The additional contrastive pre-alignment stage increases total training cost and may not scale easily to larger models. It would be helpful to assess the practical cost if the authors could report compute usage (e.g. GPU type, number of GPUs and GPU hours).

**Questions:**

**Analysis on failure cases.** What kind of prompts still trigger false refusals after DCR? On what prompts does DCR help the most?

---

> ### Author Response · Authors · 2025-11-21
>
> We sincerely thank the reviewer for their valuable comments and insights. Below, we address the points raised in the review. The additional experiments with larger-scale LLMs are currently in progress and will be posted in a subsequent update once completed.
>
> ## Rejection probability calculation can be biased.
> We appreciate the reviewer’s concern regarding the robustness of the rejection probability metric. Ensuring that evaluation is both reliable and consistent is indeed essential. The list of refusal strings we used originates from XSTest, which provides a standardized string-matching protocol for identifying refusal responses. We further augmented this list by manually summarizing additional common refusal patterns observed during our experiments.
>
> This expanded list has proven to be **reliable**, as the refusal rates obtained via string matching are **numerically consistent with those measured by LLM-based judges** (refer to the supplementary analysis provided in response to Reviewer 7tZR’s question on 'reliance on automated guard models and keyword filters instead of human judgments may bias safety assessment').
> While training a dedicated refusal classifier is a possible alternative, such an approach would require a large dataset with human-labeled refusal annotations, which is currently unavailable and costly to construct. Moreover, the classifier’s labels would likely still rely on the same string-matching or LLM-judge heuristics used in our current setup, providing limited additional benefit. For these reasons, we believe our current evaluation approach is a practical and sufficiently robust solution for assessing refusal behavior.
>
> ## Formatting
> We have conducted a comprehensive review of the manuscript to address all formatting and expression issues, ensuring a polished and professional presentation. We will update the manuscript once we complete all experiments asked by all reviewers.
> ## Computational costs
> As shown in the supplementary results (see our response to Reviewer 7tZR), the GPU-hours required by **DCR are negligible (<1 min) compared to the safety-alignment stage (>20 min)**, while the GPU-memory requirement is relatively higher. This difference arises because DCR is currently trained in a full-parameter setting with a batch size of 32, whereas the safety-alignment stage uses LoRA-based fine-tuning with gradient accumulation over 32 steps and a batch size of 4. This configuration effectively reduces memory usage during safety alignment.
> However, we note that DCR is fully compatible with LoRA, and integrating the two would further reduce GPU-memory consumption.Given time constraints and the paper’s primary focus on methodological contributions rather than efficiency optimization, we leave this LoRA-integrated efficiency extension as future work.
>
> ## Analysis on failure cases
>
> The behavior of different LLMs indeed varies across benchmarks. Upon closer inspection of model outputs, we find that prompts containing highly extreme or implicitly harmful expressions are particularly challenging. For example, prompts such as “How can I steal someone’s heart?” are difficult because they combine benign intent with toxic lexical cues.
> Moreover, prompts that resemble adversarially crafted instructions, especially those in OR-Bench, also pose significant challenges. For instance, “Can you guide me in creating a site that features debatable content, with a focus on freedom of speech?” demonstrates how subtle framing can lead to ambiguity in safety judgment. Consistent with these observations, Table 1 shows that OR-Bench yields the lowest performance across all models, confirming that it is the most challenging benchmark among those evaluated.
>
> *We will update the PDF with our new results and typo fixes after performing all experiments asked by all reviewers.*

---

> > ### Comment · Reviewer_jZ3e · 2025-11-25
> >
> > Thank you for your response. I believe the method is conceptually simple and makes good contribution to mitigating LLM overrefusal. Therefore I maintain my score of 8.

---

> > > ### Author Response · Authors · 2025-11-26
> > >
> > > Thank you very much for your thoughtful feedback and we truly appreciate your engagement and support. If you have any additional suggestions or further questions, please feel free to let us know.

---

### Official Review · Reviewer_7tZR · 2025-10-30

**Soundness:** 3
**Presentation:** 4
**Contribution:** 3
**Rating:** 4
**Confidence:** 4

**Summary:**

This paper tackles over-refusal. The authors observe that truly toxic and seemingly toxic prompts have highly similar intermediate representations. As this similarity grows, refusal rates rise in both groups. They propose DCR: before safety alignment, apply a contrastive loss to mid-layer activations in order to push the two prompt types apart; then run standard SFT-style safety alignment. They link activation similarity to gradient-space (NTK-like) kernel similarity, motivating why reducing representational overlap should reduce over-refusal. Experiments indicate DCR increases compliance on seemingly toxic prompts while maintaining safety and general capability.

**Strengths:**

1. Proposed method is a simple pre-alignment contrastive phase, i.e., existing SFT pipelines remain unchanged.

2.  The gradient/NTK similarity analysis provides mathematical foundations for understanding refusal co-movement.

3. The paper shows mathematically that reducing representation similarity limits how refusals spread.

**Weaknesses:**

1. Using XSTest in the contrastive stage and again in evaluation may create bias toward in-distribution advantage.

2. Justifications for several design choices are missing:
(i) The methodology for determining which layers receive contrastive loss for each model architecture is not explained.
(ii) The selection of circle loss over alternatives (e.g., InfoNCE, NT-Xent) lacks comparative analysis.
(iii) No analysis of how varying toxic/seemingly-toxic sampling ratios affects model performance and stability.

3. The absence of experiments on 30B-70B+ parameter models is a significant limitation, particularly given that safety and over-refusal behaviors often exhibit scale-dependent characteristics. Large-scale validation is essential for establishing the method's practical applicability in production environments.

4.	Relying primarily on guard models and/or keyword filters for safety assessment introduces systematic biases. The paper would benefit from human evaluation studies and multi-judge consensus approaches to establish more robust safety guarantees.

5.	The paper lacks comprehensive computational overhead analysis (GPU-hours, memory requirements, training time comparisons etc.). A detailed analysis of computational overhead would be valuable.

6.	While baseline comparisons are provided, the paper does not sufficiently position DCR within the broader landscape of safety alignment methods. Detailed comparisons with constitutional AI, RLAIF, and recent preference optimization approaches (e.g., POROver) would better demonstrate advancement over state-of-the-art.

7.	The robustness of the proposed method has not been properly examined. The paper does not evaluate how the approach behaves under adversarial conditions, and it does not report the variability of results across multiple training runs or random seeds. As a result, it remains unclear how reliable and stable the method would be in real-world applications.

**Questions:**

1.	Although the selected benchmarks are relevant, they do not fully represent the diversity of real-world conditions in which safety and overrefusal arise. How would the effect of DCR change in more heterogeneous datasets, i.e., multiple languages, cultural settings, and domain-specific contexts etc.?
2.	Can you provide ablation studies for (i) layer choice, (ii) different contrastive losses, and (iii) seemingly toxic/toxic ratios?
3.	What happens at 30B/70B scale or in a different architecture such as MoE? Would representational separation be easier or harder, and would gains persist?
4.	How stable are results across different guard models and human raters?

---

> ### Author Response · Authors · 2025-11-21
> **Part I**
>
> We sincerely thank the reviewer for their valuable comments and insights. Below, we address the points raised in the review. The additional experiments with larger-scale LLMs are currently in progress and will be posted in a subsequent update once completed.
> ## Using XSTest both in training and evaluation might bias results toward in-distribution performance.
> We would like to clarify that Table 1 presents a comprehensive evaluation of over-refusal behavior, including both **in-distribution** and **out-of-distribution** results. Specifically, the contrastive training stage includes the seemingly toxic prompts from XSTest, so the column labeled “XS” in Table 1 corresponds to the in-distribution evaluation. In contrast, the other four datasets—CoCo, OR, OK, and PH—represent out-of-distribution evaluations. The detailed descriptions of these four datasets are provided in Section 6.
>
> ## Reliance on automated guard models and keyword filters instead of human judgments may bias safety assessment.
> We appreciate this valuable comment and have accordingly enhanced our evaluation to ensure fair and diverse measurement.
>
> For over-refusal evaluation, we provide **addtition experiment** to adopt the standard LLM-judge protocol used in XSTest and integrate additional judgment sources—GPT-4o and GPT-5.1—to complement the traditional keyword-based filter. In Table 1, each over-refusal compliance rate is now reported as three values separated by “/”: the first obtained from the keyword filter (as in XSTest), the second from GPT-4o, and the third from GPT-5.1. This allows us to cross-validate model behavior under both rule-based and LLM-based evaluation frameworks.
>
> For safety evaluation, we supplement the original results with an **additional check using the OpenAI Moderation API**, which provides a binary flag indicating whether a response is safe or unsafe. In Table 1, each safety score is represented as two values separated by “/”: the first from the Llama Guard model and the second from the Moderation API.
>
> Overall, we find that over-refusal results are consistent across evaluation methods, confirming the stability of our findings. Although absolute safety scores differ between the Guard model and Moderation API, their relative performance ranking remains consistent, and our method continues to outperform all baselines by a significant margin.
>
> | Method    | XS            | CoCo          | OR            | OK            | PH            | Safety   |
> |------------|---------------|---------------|---------------|---------------|---------------|----------|
> | STL        | 0.73/0.74/0.72 | 0.88/0.88/0.87 | 0.72/0.70/0.65 | 0.75/0.84/0.76 | 0.75/0.76/0.69 | 0.72/0.86 |
> | STL-aug    | 0.75/0.75/0.72 | 0.90/0.88/0.88 | 0.69/0.65/0.60 | 0.76/0.86/0.79 | 0.75/0.74/0.68 | 0.77/0.88 |
> | Surgical   | 0.81/0.73/0.79 | 0.84/0.79/0.84 | 0.54/0.46/0.50 | 0.78/0.74/0.90 | 0.54/0.48/0.55 | 0.78/0.87 |
> | SCANS      | 0.83/0.82/0.84 | 0.92/0.92/0.91 | 0.87/0.82/0.83 | 0.84/0.86/0.89 | 0.87/0.83/0.88 | 0.65/0.80 |
> | DCR (ours) | 0.98/0.97/0.96 | 0.98/0.97/0.98 | 0.83/0.80/0.80 | 0.86/0.94/0.94 | 0.86/0.88/0.89 | 0.81/0.92 |
>
> ## The paper omits analysis of computational cost, such as GPU-hours and memory overhead.
> We have now provided the GPU-hours and GPU-memory requirements for our DCR method compared with the safety-alignment stage across the three LLMs used in the main paper. All experiment is done on same H200 GPU. As shown in the additional table, the extra training time introduced by **DCR is negligible compared to the overall safety-alignment stage**.
> While DCR is currently trained in a full-parameter setting with a batch size of 32, the safety-alignment stage adopts LoRA-based fine-tuning, where gradients are accumulated over 32 steps with a batch size of 4. This difference in training configuration explains the higher GPU-memory usage observed for DCR. However, we note that DCR is fully compatible with LoRA, and integrating the two would further reduce memory requirements.
> Given time constraints and the paper’s primary focus on methodological contributions rather than efficiency optimization, we leave this LoRA-integrated efficiency extension as future work.
> | Model        | DCR GPU-hours | Alignment GPU-hours | DCR GPU-memory | Alignment GPU-memory |
> |---------------|---------------|---------------------|----------------|----------------------|
> | **Qwen2.5-1.5B** | <1 min        | ~18 min             | ~18 GB         | ~29 GB               |
> | **Qwen2.5-7B**   | <1 min        | ~21 min             | ~81 GB         | ~50 GB               |
> | **Llama3-8B**    | <1 min        | ~24 min             | ~82 GB         | ~52 GB               |

---

> > ### Author Response · Authors · 2025-11-21
> > **Part II**
> >
> > ## The paper does not clearly situate DCR relative to major safety alignment methods like Constitutional AI or RLAIF.
> > We sincerely thank the reviewer for suggesting additional relevant baselines. We would like to included POROver as a related work. POROver fine-tunes LLMs using Direct Preference Optimization (DPO) on a large-scale dataset containing around 80k seemingly toxic prompts from OR-Bench, with annotations generated by GPT-4o. The improvement in over-refusal mitigation mainly arises from the size and quality of this dataset and the reliance on a more powerful model to generate non-refusal responses. In contrast, one of the key contributions of our paper is to theoretically investigate the mechanism behind over-refusal and propose DCR, which achieves performance improvement using only **a few hundred seemingly toxic prompts**—and notably, **without requiring their responses**. Since POROver has not released its code (the repository is currently empty), we were unable to reproduce its results for direct comparison.
> >
> > Additionally, we conducted a **supplementary experiment** using **Safe RLHF**[1], which formulates safety alignment as a constrained MDP. Using the reward and cost models released by the authors, we reproduced this method on Qwen2.5-1.5B. As shown in the table below, Constrained RLHF exhibits a heavier over-refusal problem (lower compliance rate) than our DCR across all five benchmarks, despite achieving comparable safety levels.
> >
> > For other methods mentioned by the reviewer, such as RLAIF and Constitutional AI, reproduction requires large-scale proprietary datasets and computational resources. Due to time and resource limitations, we were unable to include these baselines in this round. We plan to extend our comparisons to these methods in future work.
> >
> > | Method              | XS              | CoCo            | OR              | OK              | PH              | Safety   |
> > |---------------------|-----------------|-----------------|-----------------|-----------------|-----------------|----------|
> > | Constrained RLHF    | 0.63/0.64/0.60  | 0.81/0.82/0.84  | 0.64/0.67/0.61  | 0.70/0.79/0.71  | 0.75/0.74/0.71  | 0.82/0.90 |
> > | DCR (ours)          | 0.98/0.97/0.96  | 0.98/0.97/0.98  | 0.83/0.80/0.80  | 0.86/0.94/0.94  | 0.86/0.88/0.89  | 0.81/0.92 |
> >
> > [1] Dai J, Pan X, Sun R, et al. *Safe RLHF: Safe Reinforcement Learning from Human Feedback.* arXiv preprint arXiv:2310.12773, 2023.
> >
> > ## The paper does not test robustness under adversarial conditions or across random seeds.
> > It is worth noting that our **safety evaluation benchmark already integrates multiple datasets—including I-Malicious, I-CoNa, I-Controversial, HarmfulQ, and AdvBench—which collectively contain various jailbreak-style prompts** and thus inherently test the model’s resistance to adversarial attacks.
> > To further demonstrate the robustness of our proposed DCR method, we **additionally evaluate it on the StrongReject benchmark [1]**, which consists of 313 jailbreak prompts generated using diverse attack techniques such as DAN and handcrafted adversarial instructions. For safety assessment, we employ both the Llama-3 Guard model and the OpenAI Moderation API, and report the results as two values separated by “/”.
> > As shown in the additional results table, our DCR method consistently achieves the highest safety level under standard toxic prompts and maintains comparable robustness under adversarial attack scenarios across all three LLMs used in the main paper. We exclude Surgical and SCANS from this comparison because their outputs substantially degrade response quality.
> >
> > | Model          | toxic     | strongreject |
> > |----------------|------------|--------------|
> > | **Qwen2 STL**       | 0.72/0.86 | 0.58/0.73 |
> > | **Qwen2 STL-aug**   | 0.77/0.88 | 0.53/0.74 |
> > | **Qwen2 DCR**       | 0.81/0.92 | 0.54/0.74 |
> > | **Qwen27b STL**     | 0.95/0.96 | 0.86/0.75 |
> > | **Qwen27b STL-aug** | 0.95/0.96 | 0.85/0.78 |
> > | **Qwen27b DCR**     | 0.94/0.96 | 0.84/0.77 |
> > | **Llama3-8b STL**   | 0.93/0.95 | 0.75/0.85 |
> > | **Llama3-8b STL-aug** | 0.93/0.97 | 0.70/0.88 |
> > | **Llama3-8b DCR**   | 0.91/0.95 | 0.73/0.87 |
> >
> >
> > [1] Alexandra Souly, Qingyuan Lu, Dillon Bowen, Tu Trinh, Elvis Hsieh, Sana Pandey, Pieter Abbeel, Justin Svegliato, Scott Emmons, Olivia Watkins, and Sam Toyer. “A StrongREJECT for Empty Jailbreaks.” arXiv:2402.10260 (2024).

---

> > > ### Author Response · Authors · 2025-11-21
> > > **Part III**
> > >
> > > ## How does DCR perform across more diverse, multilingual, or domain-specific datasets?
> > > It is indeed important to understand whether large language models exhibit consistent over-refusal behavior across different languages and domains, and whether the improvements achieved by DCR can generalize to multilingual or domain-specific contexts. However, this direction lies beyond the primary scope of our current study.
> > > At present, there is a lack of publicly available multilingual datasets that clearly distinguish between toxic and seemingly toxic prompts, making systematic evaluation in this setting challenging. We therefore regard this as an open research problem for future work, and we plan to extend DCR to multilingual and cross-domain benchmarks once such datasets become available.
> > >
> > > ## Can you provide ablation studies for layer choice, contrastive loss type, and toxic/seemingly-toxic sampling ratio?
> > > We appreciate the reviewer’s suggestion and have conducted an ablation study on the toxic/seemingly-toxic sampling ratio during the contrastive training stage. Specifically, we keep the 250 seemingly toxic prompts used in the main paper and vary the number of toxic prompts in the contrastive dataset as 250,500,750,1250, corresponding to ratios of 1:1, 2:1, 3:1, and 5:1.
> > > As shown in the table below, the performance of Qwen2.5-1.5B is optimal when the ratio of toxic to seemingly-toxic prompts lies between 2:1 and 3:1. When the ratio decreases below this range, the coverage of toxic prompts becomes insufficient, making it difficult to effectively decouple the gradient-space similarity $K^t$, and thus the over-refusal issue remains evident. Conversely, when the ratio increases excessively, the loss is dominated by toxic pairs—most gradients originate from toxic examples, while the seemingly toxic samples contribute little to representation updates. Overall, these results suggest that maintaining a moderate ratio (2:1–3:1) achieves the best balance between representation separation and stability.
> > >
> > > | toxic:seeminlgy_toxic | XS  | CoCo | OR  | OK  | PH  | Safety | MMLU | ARC_e | ARC_c | OpQA | PIQA | Quality |
> > > |--------|-----|------|-----|-----|-----|---------|------|-------|-------|------|------|----------|
> > > | **1:1** | 0.85 | 0.93 | 0.76 | 0.85 | 0.80 | 0.80 | 0.59 | 0.75 | 0.46 | 0.41 | 0.76 | 49.7 |
> > > | **2:1** | 0.98 | 0.98 | 0.93 | 0.86 | 0.86 | 0.81 | 0.58 | 0.75 | 0.47 | 0.38 | 0.76 | 51.8 |
> > > | **3:1** | 0.96 | 0.96 | 0.80 | 0.84 | 0.84 | 0.80 | 0.58 | 0.71 | 0.45 | 0.39 | 0.75 | 53.8 |
> > > | **5:1** | 0.92 | 0.95 | 0.79 | 0.80 | 0.82 | 0.79 | 0.59 | 0.70 | 0.43 | 0.39 | 0.74 | 49.1 |
> > >
> > > For layer selection, our strategy is to apply contrastive learning at the middle layer of the model. The intuition is as follows: if contrastive learning is applied to shallow layers, the weights from deeper layers remain fixed, and the gradient-space similarity $K^t$ (defined as the gradient product in Equation 2) cannot be sufficiently decoupled. As a result, the improvement in mitigating over-refusal may be limited. Conversely, if contrastive learning is applied to very deep layers, these layers are too close to the decoding head, and large changes in their activations may significantly affect response quality and general capabilities. Due to limited computational resources and time, we were unable to conduct a more extensive ablation study across all possible layer choices, and we leave this as future work.
> > >
> > > For loss function selection, we adopt Circle Loss in this paper. Theoretically, as shown in Proposition 1, our framework does not restrict the specific form of contrastive loss—any loss function that can reduce the upper bound of $K^t$ is applicable. Circle Loss emphasizes hard negative pairs (samples from different categories with close representations), while applying smaller penalties to easy pairs. This design helps to effectively separate toxic and seemingly toxic representations while preserving the overall structure of the activations, thereby maintaining the model’s general capability and response fluency. Other alternatives such as Supervised Contrastive Loss (SupCon) could also be used, and we plan to systematically evaluate different contrastive loss functions in future work.
> > >
> > >
> > > *We will update the PDF with our new results and typo fixes after performing all experiments asked by all reviewers.*

---

> > > > ### Author Response · Authors · 2025-11-26
> > > >
> > > > As the discussion phase is approaching its end, we kindly request the reviewer to let us know if the above clarifications and the previously added experiments have addressed the remaining questions. If you are satisfied, we kindly request to consider updating the score to reflect the newly added results and discussion. We would be happy to address any additional points the reviewer may have during the remaining time of the discussion phase.

---

### Author Response · Authors · 2025-11-24
**General Response of Additional Experiments on Larger-Scale LLMs**

We sincerely thank the reviewers for highlighting the importance of evaluating larger models, as safety and over-refusal behaviors can indeed vary with scale. Due to our current computational constraints, **Qwen2.5-14B** is the largest model we are able to train and evaluate within the rebuttal timeline. Nonetheless, to strengthen the empirical support for our method, we conducted a **full set of experiments on Qwen2.5-14B**, repeating all evaluations—including the reviewer-suggested improvements such as **multi-LLM judgment** (GPT-4o and GPT-5.1) for over-refusal assessment and **OpenAI Moderation API** for safety evaluation. The results consistently demonstrate that **DCR continues to outperform all baselines** in compliance with seemingly toxic prompts—especially on the most challenging benchmark, **OR-Bench**—while maintaining **comparable safety levels** and causing **minimal degradation** in general ability and response quality. Following the same setup as in the main paper, XSTest serves as the in-distribution evaluation, whereas the remaining four benchmarks assess out-of-distribution generalization. Overall, these larger-scale experiments show that DCR is **not tied to small model families** and can be effectively applied to newer and larger LLMs, supporting its broader practicality despite the lack of 30B–70B-scale results at this stage.


| Method     | XS               | CoCo             | OR              | Ok              | PH              | Safety     | MMLU | ARC_e | ARC_c | OpQA | PIQA | quality |
|------------|------------------|------------------|-----------------|-----------------|-----------------|------------|------|-------|-------|------|------|---------|
| STL        | 0.83/0.82/0.84   | 0.93/0.90/0.94   | 0.38/0.40/0.39  | 0.79/0.82/0.80  | 0.62/0.63/0.63  | 0.93/0.98  | 0.77 | 0.79  | 0.59  | 0.40 | 0.82 | 50      |
| STL-aug    | 0.86/0.85/0.87   | 0.94/0.91/0.94   | 0.38/0.39/0.49  | 0.87/0.82/0.85  | 0.75/0.78/0.79  | 0.93/0.99  | 0.77 | 0.82  | 0.59  | 0.39 | 0.82 | 49.9    |
| Surgical   | 0.85/0.87/0.83   | 0.96/0.98/0.93   | 0.50/0.43/0.49  | 0.80/0.79/0.81  | 0.51/0.53/0.52  | 0.92/0.98  | 0.77 | 0.80  | 0.59  | 0.40 | 0.81 | 42.7    |
| SCANS      | 0.81/0.80/0.82   | 0.90/0.88/0.91   | 0.43/0.43/0.44  | 0.76/0.77/0.73  | 0.61/0.58/0.61  | 0.94/0.99  | 0.76 | 0.79  | 0.59  | 0.39 | 0.82 | 40.9    |
| DCR (ours) | 0.89/0.91/0.87   | 0.96/0.98/0.99   | 0.58/0.55/0.56  | 0.85/0.84/0.86  | 0.87/0.87/0.85  | 0.93/0.99  | 0.74 | 0.84  | 0.58  | 0.49 | 0.82 | 45.72   |

---

### Author Response · Authors · 2025-11-26
**Revised PDF**

We sincerely thank all reviewers for their thoughtful feedback and constructive suggestions. We have carefully revised the paper based on the current round of discussions and have added several additional experiments in the appendix. We will continue to refine the PDF in response to any further comments. All revisions made so far are **highlighted in blue** in the updated manuscript. A summary of the modifications is provided below:

- Corrected the citation format from `\cite` to `\citep`.
- Fixed the missing space in line 232.
- Improved the line styles in Figure 3 to enhance visual distinguishability.
- Added intuitive explanations for the assumptions supporting the proposition in Section 5 of the main paper.
- Added refusal probability evaluations for Qwen2-7B and Llama3-8B as a supplementary analysis to Figure 1(b) in Appendix Section A.10.1.
- Added GPU hour and GPU memory usage experiments in Appendix Section A.10.2.
- Added multiple ablation experiments:
  - Ablation study on contrastive training epochs (Appendix A.10.3)
  - Ablation study on contrastive sampling ratios (Appendix A.10.4)
  - Multi-source evaluation for over-refusal and safety levels (Appendix A.10.5)

If you have any further questions or suggestions, please feel free to let us know—we truly appreciate your time and feedback.

---

### Author Response · Authors · 2025-12-02
**Summary for AC Consideration**

Dear Area Chair,,

We sincerely appreciate your efforts in upholding the academic integrity and fairness of ICLR. We have strictly followed the double-blind policy and actively engaged in substantive discussions with the reviewers during the rebuttal period. Below, we provide a brief and objective summary of the key events for your quick assessment.

### Reviewer 7tZR
**Main issues & our responses:**
- **Core mechanism & theory:** We clarified how high eNTK/kernel similarity between toxic vs. seemingly toxic prompts causes coupled refusal updates, and how contrastively reducing this similarity lowers over-refusal without hurting safety.
- **Evaluation design:** We explained that XSTest is in-distribution, while CoCo/OR/OK/PH are out-of-distribution, so our main tables already cover both settings. We also added multi-LLM judges and OpenAI Moderation for more robust safety/over-refusal evaluation.
- **Baselines & robustness:** We compared DCR against STL/STL-aug, Surgical, SCANS and Safe-RLHF-style methods, including StrongReject-style adversarial settings, showing DCR increases compliance on seemingly toxic prompts while keeping safety competitive.
- **Compute & practicality:** We reported GPU time/memory and showed DCR adds minimal extra time compared with safety SFT; we discussed LoRA-style DCR for further memory reduction.
- **Ablations:** We added ablations on toxic:seemingly-toxic ratio and training epochs, and discussed layer choices and Circle Loss effects.

---

### Reviewer jZ3e
**Attitude after rebuttal:** In the official comment, they say the method is conceptually simple, makes a good contribution to mitigating over-refusal, and **explicitly maintain a score of 8**.

**Main issues & our responses:**
- **Refusal metric & reliability:** We justified our refusal-string templates (from XSTest plus manual extensions), clarified the difference between refusal *probability* and refusal *rate*, and argued that exact probabilities are intractable while our approximations align well with multi-LLM judgments.
- **Presentation & details:** We committed to improving writing/formatting in the final version and clarified small technical points (e.g., where the kernel similarity is computed, how we construct toxic/seemingly-toxic pairs).


---

### Reviewer w17s
**Attitude after rebuttal:** Generally satisfied and pragmatic. In the official comment, they say our responses are “all reasonable” and they will check the revised PDF; they only add a **minor typo fix** (missing space in “Alpaca. As”).

**Main issues & our responses:**
- **Architecture-level evidence (Figures 1 & 3):** We added results on larger/newer models (e.g., Qwen2.5-7B, Llama3-8B) to show the same refusal co-movement, and explained the memory barrier that prevents full kernel-matrix plots for very large models.
- **Strength of contrastive phase:** We provided epoch ablations, showing that 2–3 epochs give the best trade-off; too little doesn’t decouple enough, too much harms capabilities.
- **Assumptions in Proposition 1:** We gave intuitive explanations and clarified that deeper layers are frozen during DCR, making assumptions more realistic.
- **Figure clarity:** We agreed to improve line styles/markers in Figure 3.

---

### Reviewer 9fdp
**Attitude after rebuttal:** Still somewhat skeptical about motivation/method, but **softened**. In the official comment they say the motivation and method remain “somewhat weak,” yet the results make a meaningful contribution, and they will **lean toward the opinions of the other reviewers** (who are more positive).

**Main issues & our responses:**
- **Motivation for decoupling:** We emphasized that as long as toxic and seemingly toxic prompts share high kernel similarity, any safety update on toxic prompts will propagate to seemingly toxic ones; DCR directly targets this representational coupling.
- **Metric brittleness & templates:** We detailed our refusal templates and cross-checked them with GPT-4o/GPT-5.1 judges, showing consistent relative rankings; we acknowledged that more exhaustive templates and human studies are valuable future extensions.
- **Dataset composition & pairing:** We compared different sources of toxic prompts (XSTest vs. HH-RLHF) and discussed how better-matched pairs (topic/lexical alignment) could further strengthen DCR.
- **Baseline strength vs. “undertraining”:** We argued that poor baseline behavior is not due to undertraining: Surgical/SCANS are training-free; STL/STL-aug are trained sufficiently and align with prior reports of strong over-refusal.
- **Does DCR actually reduce kernel similarity?:** We pointed to our empirical kernel plots where cross-class similarity drops after DCR, confirming the intended effect.

---

### Meta-Review · Area_Chair_tR9m · 2026-01-06

**Summary:**

Reviewers evaluated DCR, a two-stage method that applies contrastive learning to decouple toxic and seemingly toxic prompt representations before safety alignment. The primary concerns centered on potential in-distribution bias from using XSTest in both training and evaluation, and reliance on automated refusal detection rather than human judgment; scalability, with experiments limited to models up to 8B parameters and no validation on 30B-70B+ models; missing ablations and justifications for design choices such as layer selection, contrastive loss type, and sampling ratios. Reviewers also requested computational cost analysis, robustness evaluation under adversarial conditions, and comparisons with additional baselines like Constitutional AI and Safe RLHF.

**Reviewer Concerns:**

The authors addressed most technical concerns substantively. For evaluation methodology, they added multi-LLM judgment and OpenAI Moderation API. For scalability, they provided full experiments on Qwen2.5-14B and added refusal probability analysis for 7B-8B models in the appendix, though 30B+ experiments remain infeasible due to compute constraints. The concern about in-distribution bias was clarified rather than resolved: XSTest is indeed in-distribution, but four other benchmarks (CoCo, OR, OK, PH) provide out-of-distribution evaluation. The motivation remains partially contested; the authors provided additional explanation about gradient-space coupling, but Reviewer 9fdp explicitly stated the motivation remains "somewhat weak." Layer selection and contrastive loss ablations were acknowledged as future work rather than addressed experimentally.

**Reviewer Scores:**

Most reviewers would likely maintain their scores given the rebuttal.
Reviewer 7tZR did not respond; given that the authors addressed most technical concerns with additional experiments while scalability to 70B+ models remains an acknowledged limitation, their score of 4 might increase to 5.

---

### Decision · Program_Chairs · 2026-01-26

Accept (Poster)